# Attaining Human's Desirable Outcomes in Indirect Human-AI Interaction via Multi-Agent Influence Diagrams

## Abstract

In human-AI interaction, one of the cutting-edge research questions is how AI agents can assist a human to attain their desirable outcomes. Most related work investigated the paradigm where a human is required to physically interact with AI agents, which we call direct human-AI interaction. However, this paradigm would be inapplicable when the scenarios are hazardous to humans, such as mine rescue and recovery. To alleviate this shortcoming, we consider indirect human-AI interaction in this paper. More detailed, a human would rely on some AI agents which we call AI proxies to interact with other AI agents, to attain the human's desirable outcomes. We model this interactive process as multi-agent influence diagrams (MAIDs), an augmentation of Bayesian networks to describe games, with Nash equilibrium (NE) as a solution. Nonetheless, in a MAID there may exist multiple NEs, and only one NE is associated with a human's desirable outcomes. To reach this optimal NE, we propose pre-strategy intervention which is an action to provide AI proxies with more information to make decision towards a human's desirable outcomes. Furthermore, we demonstrate that a team reward Markov game can be rendered as a MAID. This connection not only interprets the successes and failures of prevailing multi-agent reinforcement learning (MARL) paradigms, but also underpins the implementation of pre-strategy intervention in MARL. In practice, we incorporate pre-strategy intervention into MARL for the team reward Markov game to model the scenarios where all agents are required to achieve a common goal, with partial agents working as AI proxies to attain a human's desirable outcomes. During training, these AI proxies receive an additional reward encoding the human's desirable outcomes, and its feasibility is justified in theory. We evaluate the resulting algorithm ProxyAgent in benchmark MARL environments for teamwork, with additional goals as a human's desirable outcomes.

## 1 Introduction

In human-AI interaction, the research questions are focused on how AI agents can assist a human to attain their desirable outcomes (Dash et al., 2023; Niszczota & Abbas, 2023; Wang et al., 2023), and ultimately how AI agents can provide societal benefits in manufacturing, healthcare, and financial decision-making (Amershi et al., 2019; Wu et al., 2021; Yang et al., 2020). However, most of these works belong to direct human-AI interaction where a human is required to physically interact with AI agents, which may not be applicable to scenarios which may be hazardous to humans such as mine rescue and recovery (Murphy et al., 2009), and humans are not allowed to physically join such as remote-controlled interventional surgical robots (Wang et al., 2010). In this paper, we consider indirect human-AI interaction, where a human relies on some AI agents which we call AI proxies to convey their intentions, and interact with other AI agents (see Definition 1.1).

**Definition 1.1.** Human-AI interaction can be categorized into two following types:

(1) Direct interaction, where a human and AI agents physically interact in an environment;

(2) Indirect interaction, where a human would rely on some AI agents which we call AI proxies, to interact with other AI agents[1] in an environment.

One promising approach to address the indirect human-AI interaction is modelling this process as a game-theoretical model, and it would be particularly interpretable if a Nash Equilibrium (NE) of the game can be aligned to a human's desirable outcomes, referred to as the optimal NE. Specifying the optimal NE is a challenging problem since there almost always exist multiple NEs in a game model. Related work has been explored under the term Nash equilibrium selection problem (Harsanyi et al., 1988) and Pareto optimality (Pardalos et al., 2008), to decide on a specific NE. Nonetheless, these methods encountered significant shortcomings which prevent seeking the optimal NE aligned to a human's desirable outcomes: (1) It is infeasible to get comprehensive information from a human to articulate their intentions; and (2) These AI agents are not specifically designed to assist a human to attain their desirable outcomes. To address these issues, this paper aims to design an approach which we call pre-policy intervention, which intervenes AI proxies' decision making to facilitate seeking the optimal NE and thus attaining a human's desirable outcomes.

More specifically, we model the indirect human-AI interaction as a multi-agent influence diagram (MAID) (Koller & Milch, 2003), an augmentation of Bayesian networks to describe multi-agent decision making to maximize the total utility. In MAIDs, the multi-agent decision process can be described as a directed acyclic graph, with variables (nodes) to describe decisions, contexts and utilities. The pre-strategy intervention is an action that assigns a probability measure called pre-strategy determined by a pre-policy, to newly added variables as parents of AI proxies' decision variable, which provides more information to influence their decisions, referred to as strategies. In this MAID, in addition to the utility variables indicating a common goal among agents, we introduce additional utility variables indicating human's desirable outcomes, which can only be influenced by AI proxies' strategies. Our goal is finding the optimal pre-policy, so as to reach the optimal NE induced by the total utility variables.

**Contribution Summary.** The contributions of this paper are summarized as follows: (1) We consider indirect human-AI interaction in this paper, where a human would rely on AI proxies to interact with other AI agents, to attain their desirable outcomes. We model this interactive process between AI proxies and other AI agents via MAIDs introduced above, where the goal is to reach the optimal NE indicating a human's desirable outcome. (2) To mitigate the issue of multiple Nash equilibria, we propose a theoretically-guaranteed method called pre-strategy intervention. (3) We propose to leverage causal effects to measure the performance of pre-strategy intervention, which also serves as an objective function to optimize the pre-policy. (4) We show that team reward Markov games (which can simulate multi-agent teamwork) (Littman, 2001) can be rendered as MAIDs. Underpinned by this evidence, we implement pre-strategy intervention in multi-agent reinforcement learning (MARL) (a promising solution to solve team reward Markov games), referred to as ProxyAgent, but with an additional reward function encoding a human's desirable outcomes. We rigorously prove that the informed shaping reward can effectively facilitate learning the optimal pre-policy. (5) Based on the theoretical results from the perspective of MAIDs, we discuss the successes and failures of two MARL paradigms: independent learning and centralised training. (6) We evaluate ProxyAgent in Multi-Agent Particle Environment (Lowe et al., 2017) and JAX-based StarCraft Multi-Agent Challenge (Samvelyan et al., 2019; Rutherford et al., 2023), where only partial agents representing AI proxies are under pre-strategy intervention. The results confirm the effectiveness of our method.

## 2 RELATED WORK

**Environment and Mechanism Design.** Environment design involves structuring or modifying the configurations of an environment to lead agent behaviours towards a specific and desirable outcome (Zhang et al., 2009; Reda et al., 2020; Gao & Prorok, 2023). In contrast, the aim of our work is not to configure the environment directly. Rather, it focuses on intervening the agent policy by pre-strategy intervention. From the perspective of environment design, this not only devises a new

---

[1]Note that when we define AI proxies we always stand from the ego view of a human of interest. As a result, those AI agents to which human cannot convey intentions, are defined as other AI agents (or AI agents in short) in this paper. Those AI agents here can be extended to more generalized concepts such as humans and other AI proxies on behalf of other humans, as discussed in Hu & Sadigh (2023). However, to enable the problem setting as concise as possible, we do not consider these extended concepts in this paper.

paradigm, but also brings about potential novel approaches for realizing the paradigm. On the other hand, mechanism design is typically pertaining to designing a game model such that the equilibrium outcomes align to the game designer's objectives (Nisan & Ronen, 1999; Cai et al., 2013). In this paper, we focus on how to design pre-strategy intervention as a mechanism to attain a human's desirable outcomes in indirect human-AI interaction.

**Human-AI Interaction in Machine Learning.** Human-AI interaction models in machine learning have been developed for several decades. Earlier works solved this problem as by first building up a human model, such as a rule-based system (Lucas & Van Der Gaag, 1991) and a Bayesian model (Stuhlmüller & Goodman, 2014). Given the assumption of a known and well-defined human model (usually as a probabilistic model or a tree-structured model), the following works investigated how to model the human-AI interactive process, so that AI agent has potential to perceive the human's goals and better assist them, relying on the mathematical tools such as partially observable Markov decision process and dynamic programming (Çelikok et al., 2022; De Peuter & Kaski, 2023). Recently, human-AI interactions have been successfully addressed in solving the game of Diplomacy, depending on the powerful large language models (LLMs) (Meta Fundamental AI Research Diplomacy Team et al., 2022). However, these works are belonging to what we call direct human-AI interaction. In this paper, we propose to employ a MAID to model indirect human-AI interaction, which is associated with a probabilistic model under specification of a full strategy profile. In contrast to the analytic models (e.g. probabilistic models), our graphical model is easy to understand and more intuitive to design any decision rules. More recently, Hu & Sadigh (2023) proposed to use LLMs as a medium to convey human's explicit intentions to a controllable agent during training, to interact with other agents. The application of LLMs here can be treated as one approach to realize the pre-policy that conveys human's desirable outcome to AI proxies in our proposed indirect human-AI interactions, though its application range can be extended to the scenario where AI proxies interacting with other agents, including both humans and AI agents. The extended applicable range can be seen as prospect, given the success of indirect human-AI interactions.

## 3 BACKGROUND: MULTI-AGENT INFLUENCE DIAGRAMS

We now review *multi-agent influence diagram* (MAID) (Koller & Milch, 2003), which is an augmentation of the Bayesian network to describe multi-agent decision making to maximize their utility. An MAID is usually described as a tuple $\mathcal{M} = (\mathcal{I}, \mathcal{X}, \mathcal{D}, \mathcal{U}, \mathcal{G}, Pr)$. $\mathcal{I}$ is a set of agents. $\mathcal{X}$ is a set of chance variables indicating decisions of nature. Each chance variable $X \in \mathcal{X}$ is associated with a set of parents $Pa(X) \subset \mathcal{X} \cup \mathcal{D}$. $\mathcal{D} := \bigcup_{i \in \mathcal{I}} \mathcal{D}_i$ is a set of all agents' decision variables, where $\mathcal{D}_i$ is the set of agent $i$'s decision variables. For a decision variable $D \in \mathcal{D}_i$, $Pa(D)$ is the set of variables whose values is informed to agent $i$ when it selects a value of $D$. $\mathcal{U} := \bigcup_{i \in \mathcal{I}} \mathcal{U}_i$ is a set of utility variables, where $\mathcal{U}_i$ is agent $i$'s utility variable as its utility function. Note that utility variables cannot be parents of other variables. MAID defines a directed acyclic graph $\mathcal{G}$ with variables $\mathcal{V} = \mathcal{X} \cup \mathcal{D} \cup \mathcal{U}$. $Pr$ is a conditional probability distribution (CPD) defined over chance variables $X$ such as $Pr(X|Pa(X))$, and utility variables $U \in \mathcal{U}$ such as $Pr(U|\mathbf{pa})$, for each $\mathbf{pa} \in dom(Pa(U))$. Note that $Pr(U|Pa(U))$ is a Dirac function (i.e. $U$ is a deterministic function). In other words, for each instantiation $\mathbf{pa} \in dom(Pa(U))$, there is a value of $U$ that is assigned probability 1, and probability 0 to other values. To simplify the notation, $U(\mathbf{pa})$ is denoted as the value of $U$ that has probability 1 when $Pa(U) = \mathbf{pa}$. The total utility that an agent $i$ obtained from an instantiation of $\mathcal{V}$ is the sum of the values of $\mathcal{U}_i$, i.e. $\sum_{U \in \mathcal{U}_i} U(\mathbf{pa})$ where $\mathbf{pa} \in dom(Pa(U))$. An example for MAID is illustrated in Appendix 8.1.

**Decision Rule and Strategy**. An agent makes decision at variable $D$ depending on its $Pa(D)$, which is determined by a *decision rule* $\delta : dom(D(pa)) \to \Delta(dom(D))$ described in Definition 3.1. $\Delta$ indicates probability distribution space over a set. An assignment $\sigma$ of decision rules to each decision $D \in \mathcal{D}$ is called a *strategy profile*. A partial strategy profile $\sigma_{\mathcal{E}}$ is an assignment of decision rules to a subset of $\mathcal{D}$, as a restriction of $\sigma$ to $\mathcal{E}$, and $\sigma_{-\mathcal{E}}$ denotes the restriction of $\sigma$ to variables in $\mathcal{D} \backslash \mathcal{E}$. The assignment of $\sigma_{\mathcal{E}}$ to the MAID $\mathcal{M}$ induces a new MAID denoted by $\mathcal{M}[\sigma]$, and each $D \in \mathcal{E}$ would become a chance variable with the CPD $\sigma(D)$. When $\sigma$ is assigned to every decision variable in MAID, the induced MAID would become a Bayesian network with no more decision variables. This Bayesian network defines a joint probability distribution $P_{\mathcal{M}[\sigma]}$ over all the variables in $\mathcal{M}$.

**Definition 3.1** (Koller & Milch (2003)). A decision rule $\delta$ for a decision variable $D$ is a function that maps each instantiation **pa** of $Pa(D)$ to a probability distribution over $dom(D)$. An assignment of decision rules to every decision $D \in \mathcal{D}_i$ for an agent $i \in \mathcal{N}$ is called a strategy.

**Expected Utility and Nash Equilibrium.** Given a strategy profile assigned to each decision variable, with the resulting joint probability distribution $P_{\mathcal{M}[\sigma]}$ and the suppose that $\mathcal{U}_i = \{U_1, ..., U_m\}$, we can write the expected utility for an agent $i$ such that

$$\mathbb{E}U_i(\sigma) = \sum_{(u_1,...,u_m) \in dom(\mathcal{U}_i)} P_{\mathcal{M}[\sigma]}(u_1,...,u_m) \sum_{k=1}^{m} u_k. \tag{1}$$

Given Equation 1, we further define that the strategy $\sigma_{\mathcal{E}}^*$ is optimal for $\sigma$, for a subset $\mathcal{E} \subset \mathcal{D}_i$, if $\mathbb{E}U_i((\sigma_{-\mathcal{E}}, \sigma_{\mathcal{E}}^*)) \geq \mathbb{E}U_i((\sigma_{-\mathcal{E}}, \sigma_{\mathcal{E}}'))$, as shown in Definition 3.2. Furthermore, if for all agents $i \in \mathcal{I}$, $\sigma_{\mathcal{D}_i}$ is optimal for the strategy profile $\sigma$, then $\sigma$ is a Nash equilibrium, as shown in Definition 3.3.

**Definition 3.2** (Koller & Milch (2003)). Let $\mathcal{E}$ be a subset of $\mathcal{D}_i$, and let $\sigma$ be a strategy profile. $\sigma_{\mathcal{E}}^*$ is optimal for the strategy profile $\sigma$ if, in the induced MAID $\mathcal{M}[\sigma_{-\mathcal{E}}]$, where the only remaining decisions are those in $\mathcal{E}$, the strategy $\sigma_{\mathcal{E}}^*$ is optimal, for all strategies $\sigma_{\mathcal{E}}'$, such that

$$\mathbb{E}U_i((\sigma_{-\mathcal{E}}, \sigma_{\mathcal{E}}^*)) \geq \mathbb{E}U_i((\sigma_{-\mathcal{E}}, \sigma_{\mathcal{E}}')).$$

**Definition 3.3** (Koller & Milch (2003)). A strategy profile $\sigma$ is a Nash equilibrium for a MAID $\mathcal{M}$ if for all agents $i \in \mathcal{N}$, $\sigma_{\mathcal{D}_i}$ is optimal for the strategy profile $\sigma$.

For each MAID there can be multiple NEs (corresponding to multiple strategy profiles), we denote the random variable describing a possible NE over a set of NEs, $\{\hat{\sigma}_1, \ldots, \hat{\sigma}_k\}$ as $\hat{\boldsymbol{\sigma}}$. For any $\hat{\sigma} \in dom(\hat{\boldsymbol{\sigma}})$, we define the probability for an arbitrary NE as $P_{\sigma}(\hat{\sigma}) := Pr(\hat{\sigma}_{D_1}, \ldots, \hat{\sigma}_{D_i}, \ldots, \hat{\sigma}_{D_n})$, where $n := |\mathcal{N}|$ is the number of agents in the MAID. The probability of a strategy profile is defined as the joint probability that each agent $i$ plays some strategy on the agent's decision variable $D_i$.

### 3.1 RELEVANCE GRAPH

A *relevance graph* as shown in Definition 3.4 defines a directed graph describing the binary relation between two decision variables. If there exists an edge $D' \to D$, it implies that the decision variable $D$ is *strategically relies* on another decision variable $D'$. In other words, the decision rules for $D'$ is required to evaluate the decision rules for $D$. If there exist both $D' \to D$ and $D \to D'$, then the relevance graph is cyclic. Furthermore, if $D$ and $D'$ belong to two agents respectively, their payoffs depend on the decisions at both $D$ and $D'$. In this situation, the optimality of one agent's decision rule is coupled with another agent's decision rule, and the only way is to make these two agents' decision rules matched (Koller & Milch, 2003), such as choosing both agents' decision rules together, analogous to *centralised training* in multi-agent reinforcement learning (Oliehoek et al., 2008).

**Definition 3.4** (Koller & Milch (2003)). A node $D'$ is s-reachable from a node $D$ in a MAID $\mathcal{M}$ if there is some utility node $U \in \mathcal{U}_D$ such that if a new parent $\hat{D}'$ were added to $D'$, there would be an active path (Appendix 8.2) in $\mathcal{M}$ from $\hat{D}'$ to $U$ given $Pa(D) \cup \{D\}$, where a path is active in a MAID if it is active in the same graph, viewed as a Bayesian network. The relevance graph for a MAID $\mathcal{M}$ is a directed graph whose nodes are the decision nodes of $\mathcal{M}$, and which contains an edge $D' \to D$ if and only if $D'$ is s-reachable from $D$.

## 4 ATTAINING HUMAN'S DESIRABLE OUTCOMES VIA MAIDs

In this section, we outline our approach to address the core challenge of reaching the optimal NE that describes human's desirable outcomes in human-AI interaction. The overall idea centers on modelling the whole process as a game expressed in MAIDs and identifying the optimal decision rule which we refer to as pre-strategy intervention. We begin with an example that demonstrates why an agent representing a human may not always reach their desirable outcomes when interacting with other AI agents. Owing to the fact that an induced MAID $\mathcal{M}_{\sigma}$ is a causal Bayesian network, we formally define the causal effect of pre-strategy intervention, and introduce a systematic method to identify the optimal pre-strategy.

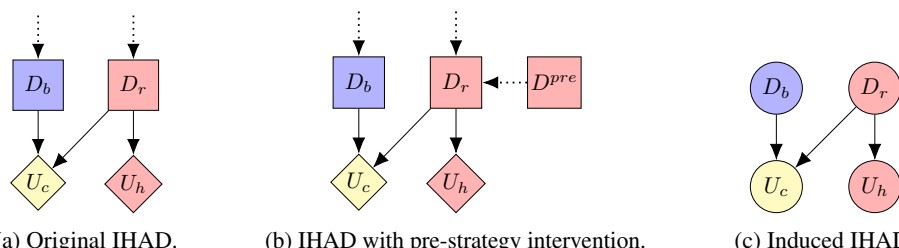

Figure 1: (a) Original IHAD, where squares indicate decision variables, diamonds indicate utility variables. The variables in red are associated with an AI proxy, while in blue associated with an AI agent. The variables in yellow ($U_c$) are associated with variables shared between agents. (b) IHAD where the AI proxy is under pre-policy intervention. (c) Induced IHAD when all decision variables are specified with strategies (possibly under pre-policy intervention), and all variables (neglecting pre-decision variables) become chance variables. Thus, the IHAD is reduced to a Bayesian network.

## 4.1 Pre-Strategy Intervention in Indirect Human-AI Diagrams

**Indirect Human-AI Diagram.** We first define the indirect human-AI diagram (IHAD), as shown in Definition 4.1, to formalize the description in Definition 1.1. To ease the understanding, we give an example in Figure 1(a), where one agent representing a human interacts with another AI agent, to not only achieve a common goal $U_c$, but also accomplish the human's desirable outcomes denoted by $U_h$. Note that only the agents representing a human agrees to maximize $U_h$. Intuitively, the additional utility variables can change the total utility as defined in Equation 1, so as to shape the optimal Nash equilibrium corresponding to the human's desirable outcomes.

**Definition 4.1.** An indirect human-AI diagram can be specified as a MAID, with specific utility variables. In details, in addition to the common utility variables $U_c \in \mathcal{U}_c \subset \mathcal{U}$ that all agents in the environment agree to maximize, utility variables $U_h$ indicating a human's desirable outcomes are added. Note that $U_h \in \mathcal{U}_h \subset \bigcup_{i \in \mathcal{H}} \mathcal{U}_i$, where $\mathcal{H} \subset \mathcal{N}$ is a set of AI proxies.

**Pre-Strategy Intervention.** To regulate the AI proxies to additionally maximize the utility variables $U_h$ indicating the human's desirable outcomes, we propose to add a *pre-decision variable* $D^{pre}$ as a new parent to a decision variable $D$ of a proxy agent, as shown in Figure 1(b). In analogy to decision variables in MAIDs, we need to give assignment $\sigma^{pre}$ which we refer to as *pre-strategy*, and this action is called *pre-strategy intervention*. This definition refers to the *stochastic intervention* defined in causal Bayesian networks (Pearl, 2009)[Chap. 4], underpinned by the fact that an induced IHAD can be treated as a causal Bayesian network, as shown in Figure 1(c). Similar to decision rules, a pre-strategy is determined by a *pre-policy* denoted by $\delta^{pre}$, as shown in Definition 4.2.

**Definition 4.2.** For a decision variable $D \in \mathcal{D}$ in a MAID, a pre-strategy intervention is an action to assign a pre-strategy $\sigma^{pre}$ to a new parent $D^{pre}$ added to $D$, referred to as pre-decision variable. The pre-strategy $\sigma^{pre}$ is determined by a pre-policy $\delta^{pre}$.

## 4.2 Navigating Rational Outcomes through Pre-Policy

Motivated by the example above, a question arises: how a pre-strategy is identified to encode a specific human's desirable outcome. First, we introduce the total utility variable, denoted as $U_{tot} := U_{tot}^h + U_{tot}^c$, where $U_{tot}^h := \sum_{U \in \mathcal{U}_h} U$ and $U_{tot}^c := \sum_{U \in \mathcal{U}_c} U$. The optimal NE is defined as $U_{tot} = u^*$. We realize this by first defining the causal effect of pre-strategy interventions on the optimal NE, and then the pre-strategy attaining the human's desirable outcomes can be identified by maximizing the causal effect.

### 4.2.1 Definition of Causal Effect of Pre-Strategy Intervention

**Definition 4.3.** Consider a pre-strategy intervention (Section 4.1) is applied to AI proxies, on its strategy profile, to influence the $U_{tot} = u^*$ induced by the optimal NE $\hat{\sigma}^*$, which is induced by a pre-strategy intervention $\sigma^{pre}$ on new AI proxies' decision rules. The set of NEs before pre-strategy

intervention is denoted by $\hat{\sigma}$. The causal effect is defined as the following equation:

$$\Delta_{\text{CE}}^{\sigma^{pre}}(U_{tot} = u^*) = \underbrace{P_{\mathcal{M}[\hat{\sigma}]}(U_{tot} = u^*)P_\sigma(\hat{\sigma}^*)}_{P_\mathcal{I}(U_{tot}=u^*)} - \underbrace{\int_{\hat{\sigma}\in\hat{\boldsymbol{\sigma}}} P_{\mathcal{M}[\hat{\sigma}]}(U_{tot} = u^*)P_\sigma(\hat{\sigma})\,d\hat{\sigma}}_{P(U_{tot}=u^*)} \quad (2)$$

In Equation 2, $P_{\mathcal{M}[\hat{\sigma}]}(U_{tot} = u^*)$ represents the likelihood of desired outcome $U_{tot} = u^*$ under the specific NE $\hat{\sigma}$. The terms $P_\sigma(\hat{\sigma}^*)$ and $P_\sigma(\hat{\sigma})$ denote the probability distributions of the optimal NE and an arbitrary NE, respectively (See Definition 3.3). The causal effect quantifies the total probabilities of $U_{tot} = u^*$ under pre-strategy intervened and original IHADs. However, it may be difficult to find the optimal pre-strategy intervention that induces the optimal NE. We prove that in this case, there exists a pre-strategy intervention maximizing the causal effect, even if the pre-strategy intervention induces a set of NEs including the optimal NE (as a weaker result), as delineated in Proposition 4.4.

**Proposition 4.4.** *Given a MAID $\mathcal{M}$, assume that the function $P_\mathcal{I}$, representing the probability of observing $U_{tot} = u^*$ under a pre-strategy intervention, is upper semicontinuous and defined on a compact domain $\text{dom}(\sigma^{pre}) \subseteq \mathbb{R}^m$. Under these conditions, there exists at least one pre-strategy of agent $i$ that does not decrease the probability of $U_{tot} = u^*$. Furthermore, there exists a pre-strategy that maximizes the causal effect.*

About the condition for Proposition 4.4 to hold, we only assume semi-continuity for the function of the probability measure $P_\mathcal{I}$ since it is usually not everywhere continuous. An intuitive example is the game *paper, rock, scissors*, where the best response is conducting each action uniformly. If we consider a pre-policy that shifts one player towards slightly less likely playing rock, then the probability of the opponent playing paper would experience a "jump" to 0, which can be seen as a discontinuity in the function. An example of pre-strategy intervention can be found in Appendix 9.

### 4.2.2 ATTAINING HUMAN'S DESIRABLE OUTCOMES BY PRE-STRATEGY INTERVENTION

Having formalized the causal effect of a pre-strategy intervention and established the existence of an optimal pre-strategy intervention that maximizes the causal effect above, a pertinent question now arises: how a pre-strategy is evaluated. Maximizing the causal effect, as defined in Equation 2, essentially involves maximizing the likelihood of $U_{tot} = u^*$ within the intervened distribution of different strategy profiles, as the second term in the equation remains constant across interventions.

To practically evaluate a generic pre-policy that generates pre-strategies, we propose the following expression:

$$P(U_{tot} = u^* \mid \text{do}(\sigma^{pre})) = \sum_{\sigma \in \boldsymbol{\sigma}} P(U_{tot} = u^* \mid \sigma)P_\sigma(\sigma \mid \text{do}(\sigma^{pre})). \quad (3)$$

where the (full) strategy profile $\sigma$ incorporates the pre-strategy $\text{do}(\sigma^{pre})$ as a condition.

In Equation 3, the first term on the RHS is the conditional probability of an outcome $U_{tot} = u^*$ under the strategy profiles, and the second term is the distribution of agents' strategies under pre-strategy intervention. This formulation implies that *we first allow agents to learn their best response strategies to each other given pre-strategies. Then, it is eligible to evaluate the likelihood of the outcome $U_{tot} = u^*$ based on the full set of strategy profiles and updating the pre-strategy accordingly.*

### 4.3 RENDERING MARKOV GAMES AS MAIDS

Markov game (Littman, 1994) is a popular mathematical model to describe the multi-agent decision process across various real-world applications (Qiu et al., 2021; Wang et al., 2021; Zhang et al., 2024). The success to associate Markov games with MAIDs, can enable implementing pre-policy intervention in multi-agent reinforcement learning (MARL), a common paradigm to solve Markov games. Furthermore, the theoretical results behind MAIDs can reversely facilitate understanding centralised training and independent learning in MARL. For succinct description, we only consider the team reward Markov game with a finite episode length, as shown in Definition 4.5.

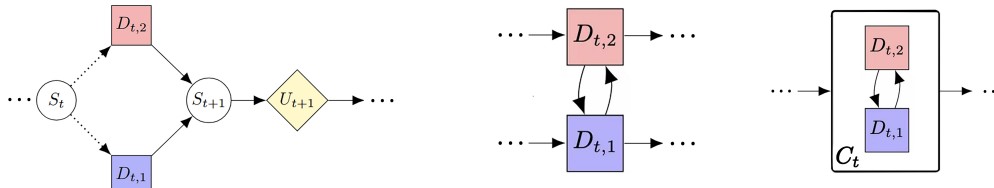

(a) MAID rendering a team reward Markov game.

(b) Relevance graph indicating independent learning.

(c) Component graph indicating centralised training.

Figure 2: Illustrations of rendering team reward Markov games as MAIDs, and relevance graphs associated with MARL paradigms. The red and blue squares indicate two agents' decision variables, respectively. The yellow utility variables indicate common utility variables shared between agents. The white squares with a bold $C_t$ indicate a maximal SCC.

**Definition 4.5** (Littman (2001)). A team reward Markov game can be described as a tuple $\langle \mathcal{N}, \mathcal{S}, \mathcal{A}, T, R, L \rangle$. $\mathcal{N}$ is a set of agents; $\mathcal{S}$ is a set of states; $\mathcal{A} = \times_{i \in \mathcal{N}} \mathcal{A}_i$ is a set of joint actions and $\mathcal{A}_i$ is agent $i$'s action set; $T : \mathcal{S} \times \mathcal{A} \to \mathcal{S}$ describes the transition function that maps a state $s_t \in \mathcal{S}$ at timestep $t$ to $s_{t+1} \in \mathcal{S}$ at timestep $t + 1$; $R : \mathcal{S} \times \mathcal{A} \to \mathbb{R}$ is a team reward function that evaluate the immediate joint action $a_t \in \mathcal{A}$ at some state $s_t \in \mathcal{S}$. In a team reward Markov game with an episode length of $L$ timesteps, agents aim to learn a joint policy $\pi = (\pi_i)_{i \in \mathcal{N}}$ where $\pi_i : \mathcal{S} \to \mathcal{A}_i$ is agent $i$'s stationary policy, to solve the following optimization problem such that $\max_\pi \mathbb{E}_{\pi,T}[\sum_{t=0}^{L} R(s_t, a_t)]$.

A team reward Markov game can be described as a directed acyclic graph. It can be rendered as a MAID as Figure 2(a) shows, because we can match variables between these two models. In details, both models' agent sets are $\mathcal{N}$; $\mathcal{S}$ is associated with chance variables $\mathcal{X}$; $\mathcal{A}$ is associated with decision variables $\mathcal{D}$; $T$ is associated with conditional probability distributions $Pr$; $\pi$ is associated with decision rules $\delta$; and $\mathbb{E}_{\pi,Pr}[\sum_{t=0}^{T} R(s_t, a_t)]$ is associated with the expected utility as shown in Equation 1. The objective of a team reward Markov game $\max_\pi \mathbb{E}_{\pi,Pr}[\sum_{t=0}^{L} R(s_t, a_t)]$ is equivalent to reaching a Nash equilibrium (see Definition 3.3), given that each agent is equipped with common utility variables, as defined in indirect human-AI diagrams, a specification of MAIDs for modelling indirect human-AI interactions (see Definition 4.1).

### 4.3.1 KEY INSIGHTS INTO MARL PARADIGMS

Having rendered a team reward Markov game as a MAID, we now give some insights into the popular MARL paradigms such as *independent learning* (Claus & Boutilier, 1998) and centralised training (Oliehoek et al., 2008), through the lens of MAIDs.

**Independent Learning.** It is not difficult to observe that the team reward Markov game is a simultaneous move game. If each agent learns independently, it would lead to an issue called *non-stationarity dilemma* (Hernandez-Leal et al., 2019). Literally, this is caused by the situation that each agent is not informed with others' decisions and independently updates its policy, with regarding other agents as part of the environment. If we express the team reward Markov game as a s-relevance graph as shown in Figure 2(b), a cycle would appear between decision variables at each timestep. As per the discussion in Section 3.1, it is not guaranteed to reach a Nash equilibrium by solely determining each agent's decision variables, with a generalized backward induction algorithm. This is in principle aligned with the *temporal-difference* (TD) learning (Sutton, 2018)[Chap. 6] and the *actor-critic* algorithms (Konda & Tsitsiklis, 1999), which underpin the modern on-policy and online algorithms for single-agent reinforcement learning. In turn, this association can well explain the failure of independent learning, as a single-agent reinforcement learning algorithm.

**Centralised Training.** Recall that the non-stationarity dilemma above can be well solved by centralised training (Oliehoek et al., 2008), which treats a team of agents as a whole executing joint actions. Thereby, Markov game is reduced to a Markov decision process as a single-agent case. This paradigm can be interpreted from the perspective of MAID, as transforming a cyclic s-relevance graph to a *component graph* with the *maximal strongly connected components* (SCCs) as nodes, as shown in Figure 2(c). More specifically, a maximal SCC includes the decision variables forming a cyclic s-relevance graph at each timestep. Koller & Milch (2003) showed that solving the acyclic

component graph [2] via the generalized backward induction algorithm can reach a Nash equilibrium. This is associated with centralised training employed to reach the maximum cumulative team rewards, as a Nash equilibrium in a team reward Markov game (Littman, 2001; Oliehoek et al., 2008).

### 4.3.2 Pre-Policy Learning

---
**Algorithm 1** ProxyAgent
---
1: Initialize $\pi_{\theta_{\text{pre}}}$ (pre-policy) and $\pi_{\theta_{\text{agent}}}$ (agents' policies)
2: Define environments $\mathcal{E}_{\text{pre}}$ (with shaping rewards) and $\mathcal{E}_{\text{norm}}$ (with extrinsic rewards)
3: **while** Pre-policy and agents' policies have not converged **do**
4:     **for** a fixed number of updates **do**                ▷ Stage 1: Updating the pre-policy
5:         Update $\pi_{\theta_{\text{pre}}}$ given $\pi_{\theta_{\text{agent}}}$ in $\mathcal{E}_{\text{pre}}$
6:     **end for**
7:     **for** a fixed number of updates **do**              ▷ Stage 2: Updating agents' policies
8:         Update $\pi_{\theta_{\text{agent}}}$ given $\pi_{\theta_{\text{pre}}}$ in $\mathcal{E}_{\text{norm}}$
9:     **end for**
10: **end while**
11: **Return:** $\pi_{\theta_{\text{pre}}}$, $\pi_{\theta_{\text{agent}}}$
---

Based on the equivalence between team reward Markov games and MAIDs, it is natural to implement pre-policy intervention in MARL. In Equation 3, we formulated the evaluation of pre-policy intervention from the perspective of causal effects, which will be the objective function in our algorithm. Before detailing our algorithm, we first justify that observing utility variables and pre-strategy intervention are instrumental in reaching the optimal NE as a human's desirable outcome. As shown in Equation 4, the summand with respect to the optimal NE of the RHS in Equation 3 is proportional to $P(\hat{\sigma}^* \mid U_{tot} = u^*, \text{do}(\sigma^{pre}))$, the posterior probability of the optimal NE $\hat{\sigma}^*$. Consequently, maximizing causal effects shown in Equation 3 is equivalent to maximum a posterior with respect to the optimal NE $\hat{\sigma}^*$. The observations of the posterior probability illuminates the necessity of observing utility variables and pre-strategy intervention to maximize the probability of reaching the optimal NE.

$$P(\hat{\sigma}^* \mid U_{tot} = u^*, \text{do}(\sigma^{pre})) \propto P_{\mathcal{M}[\sigma]}(U_{tot} = u^* \mid \hat{\sigma}^*)P_\sigma(\hat{\sigma}^* \mid \text{do}(\sigma^{pre})). \qquad (4)$$

In the context of MARL, common utility variables and the utility variables measuring a human's desirable outcomes are implemented as extrinsic rewards and intrinsic rewards (Mguni et al., 2022), respectively. As Algorithm 1 shows, Stage 1 aims at maximizing $P_\sigma(\sigma \mid \text{do}(\sigma^{pre}))$ given fixed $P_{\mathcal{M}[\sigma]}(U_{tot} = u^* \mid \sigma)$: the pre-policy is optimized with shaping rewards as the sum of intrinsic rewards encoding a human's desirable outcomes and extrinsic rewards emitted from the environment, with fixing agents' policies (decision rules $\delta$). Thus, actions (strategies $\sigma$) generated would be determined by pre-strategies $\sigma^{pre}$ generated by the pre-policy (pre-decision rules $\delta^{pre}$). Stage 2 is focused on maximizing $P_{\mathcal{M}[\sigma]}(U_{tot} = u^* \mid \sigma)$ given fixed $P_\sigma(\sigma \mid \text{do}(\sigma^{pre}))$: agents' policies are optimized with shaping rewards, with fixing the pre-policy and thus fixing pre-strategies $\sigma^{pre}$. Iterating between Stage 1 and Stage 2 is expected to result in that $\sigma \rightarrow \hat{\sigma}^*$, and $P(\hat{\sigma}^* \mid U_{tot} = u^*, \text{do}(\sigma^{pre}))$ is maximized, i.e., the human's desirable outcomes have been attained.

## 5 Experiments

The above sections show how the optimal NE associated with a human's desirable outcomes is attained using pre-policy intervention. The evaluation of Algorithm 1 is focused on answering the following two research questions: (1) Is the pre-policy able to attain a human's desirable outcomes (measured by intrinsic rewards)? (2) Would a human's desirable outcomes affect the goal of the original task (measured by extrinsic rewards)?

### 5.1 Experiments Setup

In experiments, we intervene some agents as AI proxies in the environment and these agents are fed with intrinsic rewards. All experiments are conducted with ten random seeds, and the results

---

[2]A component graph is always acyclic (Cormen et al., 2022).

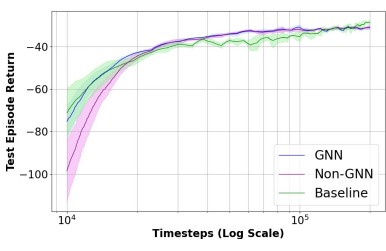

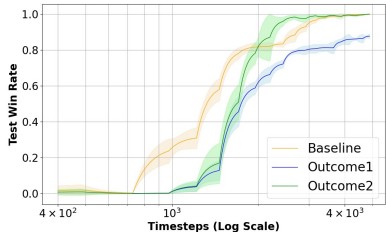

(a) Extrinsic returns for MPE.

(b) Win rate for 3s2z in SMAX.

Figure 3: Comparison between our method ProxyAgent and the baseline. Outcome 1 and Outcome 2 in (b) stands for two different human's desirable outcomes for 3s2z in SMAX.

are presented as the mean performance with a 95% confidence bar. For each test timestep, 128 episodes are evaluated. In implementation, using graph-neural networks (GNNs) (Wu et al., 2020) as a graph-based feature extraction approach is investigated, as outlined in Algorithm 2 in Appendix 12.2. The motivation is to verify the effectiveness of graph-based representation of an environment, thanks to multi-agent influence diagrams (MAIDs) we discussed in this paper.

**Multi-Agent Particle Environment (MPE).** MPE Simple Spread is a multi-agent environment where agents must cooperatively navigate to different landmarks in a 2D continuous space while avoiding collisions (Lowe et al., 2017; Rutherford et al., 2023). In our experimental setup, there exist 3 agents and 3 landmarks. There is only one AI proxy in this environment, receives intrinsic rewards to measure how far the AI proxy is to the leftmost landmark, the smaller the distance the larger intrinsic rewards.

**JAX-based StarCraft Multi-Agent Challenge (SMAX).** SMAX is a JAX-based implementation of the StarCraft Multi-Agent Challenge (SMAC), a benchmark designed for testing MARL algorithms using simplified StarCraft II combat scenarios (Samvelyan et al., 2019; Rutherford et al., 2023). We first evaluate our method in the scenario 3s2z. We design two specific cases: (1) the two Stalkers and one Zealot serve as AI proxies denoted by Outcome 1, and (2) the three Stalkers serve as AI proxies denoted by Outcome 2. In either case, AI proxies are required to form a line in attacking. We additionally evaluate our method in 3s2h and 5m_vs_6m. The AI proxies in these two scenarios are 3 Stalkers and 5 Marines, respectively. The relevant intrinsic rewards evaluate if all AI proxies stand in a line, without gathering together.

**Baseline and Ablation Variants.** All baselines share the same architecture and training setups as ProxyAgent, except for GNNs as feature extraction in ProxyAgent. For both MPE and SMAX environments, we compare the performance of our method against a baseline using standard training paradigms, such as DQN (Mnih et al., 2015) for MPE, and VDN (Sunehag et al., 2017) and PPO (Schulman et al., 2017) for SMAX. Due to page limits, we show the results of PPO in Appendix 13.1. All implementation details are provided in Appendix 12 and Appendix 14.

## 5.2 MAIN RESULTS

Figure 3 shows that our method demonstrates general faster convergence compared with the baseline. More specifically, our method can achieve high returns in MPE, while a 100% win rate in SMAX. The faster convergence implies that pre-strategy intervention actually changes the landscape of utilities and thus influences learning process. Furthermore, the difference between final returns obtained by our method and the baseline verifies that human's desirable outcomes could affect the task goal.

## 5.3 ATTAINING HUMAN'S DESIRABLE OUTCOMES

Figure 4(a) visualizes the process of the AI proxy to reach the leftmost landmark in MPE during learning. As seen from Figure 4(b), the trend of intrinsic return agrees to the changes of the AI proxy's motions. To give a more intuitive understanding about results of MPE, we conduct a case study to analyze the complexity of multiple NEs and the optimal NE in Appendix 11. Similarly, we verify the effect of pre-strategy intervention on SMAX. As shown in Figure 5(a), in all scenarios the average intrinsic reward of one episode in test demonstrates the necessity of introduce an intrinsic reward to guide reaching the optimal NE. We have noticed that it is still possible to reach the optimal

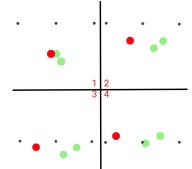 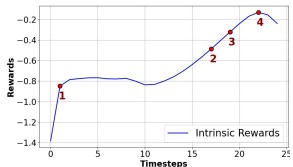

| Method | Intrinsic Reward |
|---|---|
| **Ours** | $-1.12 \pm 0.70$ |
| **Baseline** | $-1.92 \pm 0.63$ |

(a) Visualization for evaluation.    (b) Intrinsic reward per step in (a).    (c) Average last-step intrinsic rewards evaluated by 10 episodes.

Figure 4: Visualization and evaluation curve to demonstrate the intrinsic rewards across timesteps in MPE during training. The numbers from small to large in (a) indicate the sequence of subfigures visualizing the change of agents' behaviors. Furthermore, the agent in red is the AI proxy.

| Scenario | Baseline | Our Method |
|---|---|---|
| **3s2z** | $-2.0 \pm 0.42$ | $-0.7 \pm 0.11$ |
| **3s2h** | $-0.81 \pm 0.30$ | $-0.62 \pm 0.26$ |
| **5m_vs_6m** | $-0.56 \pm 0.30$ | $-0.50 \pm 0.10$ |

(a) Average intrinsic rewards evaluated by 10 episodes, across three scenarios. All rewards have been scaled by a factor of 100 for ease to demonstrate.

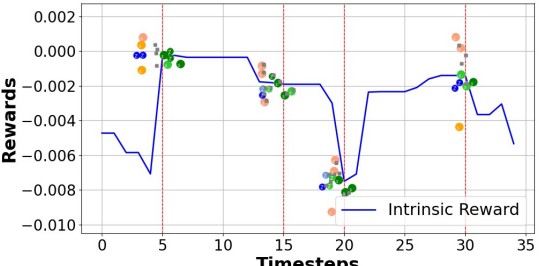

(b) Visualized Outcome 1 in 3s2z.

(c) Visualization of intrinsic rewards for Outcome 2 in 3s2z across timesteps. The agents in yellow are the AI proxies.

Figure 5: Visualization and numeric results of variant scenarios in SMAX for demonstrating intrinsic rewards across timesteps during training.

NE but the result is not controllable (only appearing once in three scenarios for baselines). The sub-optimality for Outcome 1 of 3s2z in SMAX is visualized in Figure 5(b), which we will discuss in details in Section 6. Similar to MPE, we also demonstrate a progressive visualization of how the instantaneous intrinsic reward changes for Outcome 2 of 3s2z in Figure 5(c). It can be seen that the intrinsic reward changes with the corresponding formation of AI proxies. The good performances cross different agent types and scenarios verify that our method is generally effective.

## 6   CONCLUSION, DISCUSSION AND LIMITATION

In this paper, we contributed a novel method for indirect human-AI interaction, building on the concept of pre-strategy intervention within multi-agent influence diagrams (MAIDs). Our method allows AI proxies to represent a human to interact with other AI agents, to attain their desirable outcomes but still attempt to complete the task as much as possible. The pre-strategy intervention aims to provide more information to attain the human's desirable outcomes. Based on the theory we established, we can implement pre-policy intervention in multi-agent reinforcement learning with theoretical guarantees and interpretation. We evaluate our proposed method called ProxyAgent in two benchmarks: Multi-Agent Particle Environment (Lowe et al., 2017) and JAX-based StarCraft Multi-Agent Challenge (Samvelyan et al., 2019; Rutherford et al., 2023), where only partial agents representing AI proxies are under pre-strategy intervention. The experimental results verify the effectiveness of our method and validity of our theory established on MAIDs.

**Discussion and Limitation.** As seen from Figure 3(b) and 5(b), there exists some outcome specified by intrinsic rewards which cannot be attained under some task goal specified by team rewards. This is highly dependent on the consistency between the design of intrinsic rewards and the definition of team rewards. In the future, it is a valuable research avenue to study the relation amongst the function class of intrinsic rewards, team rewards and the existence of the optimal NE to complement the framework of MAIDs. On the other hand, as Figure 3(a) shows, the effectiveness of encoding observations into graphs with pre-process by GNNs is limited, though in theory it should be more effective. The main reason could be that the graph structures we formed could deviate from the optimal structure. To remedy this issue, some research about causal discovery (Glymour et al., 2019) could be incorporated in the future.

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

# 7 NOTATION

Table 1: Summary of Notation

| Notation | Description |
|---|---|
| $\mathcal{M}$ | Multi-Agent Influence Diagram (MAID), |
| $\mathcal{I}$ | Set of agents in the MAID. |
| $\mathcal{X}$ | Set of chance variables (representing decisions of nature). |
| $\mathcal{D}$ | Set of decision variables for all agents. |
| $\mathcal{D}_i$ | Decision variables for agent $i$. |
| $Pa(D)$ | Parent set of decision variable $D$. |
| $\mathcal{U}$ | Set of utility variables. |
| $\mathcal{U}_i$ | Utility variables for agent $i$. |
| $\mathcal{G}$ | Directed acyclic graph (DAG) of the MAID. |
| $Pr$ | Conditional probability distribution (CPD) |
| $D^{pre}$ | Pre-strategy decision variable |
| $\sigma$ | Strategy profile (assignment of decision rules). |
| $\sigma^{pre}$ | Pre-strategy assigned to decision variable $D^{pre}$. |
| $\sigma_{\mathcal{E}}$ | Partial strategy profile on subset $\mathcal{E} \subseteq \mathcal{D}$. |
| $\sigma_{-\mathcal{E}}$ | Strategy profile restricted to decisions outside $\mathcal{E}$. |
| $\delta^{pre}$ | Pre-policy, which determines a pre-strategy $\sigma^{pre}$. |
| $\boldsymbol{\sigma}_{\mathcal{I}}$ | Set of strategy profiles after pre-policy intervention. |
| $U$ | Utility variable, representing human's desirable outcome. |
| $U_h$ | Utility variable indicating human's desirable outcomes. |
| $U_c$ | Common utility variable representing shared goals. |
| $P_{\mathcal{M}[\sigma]}$ | Joint probability distribution induced by strategy profile $\sigma$. |
| $\Delta_{\mathrm{CE}}(\boldsymbol{\sigma}_{\mathcal{I}}, U = u)$ | Causal effect of pre-strategy intervention $\boldsymbol{\sigma}_{\mathcal{I}}$ on outcome $U = u$. |
| $P_{\mathcal{M}[\sigma]}(U = u)$ | Likelihood of outcome $U = u$ under strategy profile $\sigma$. |
| $P_\sigma(\sigma)$ | Probability distribution over strategy profiles. |
| $\mathbb{E}U_i(\sigma)$ | Expected utility for agent $i$ under strategy profile $\sigma$. |
| $Pr(\hat{\sigma}_{D_1}, \ldots, \hat{\sigma}_{D_n})$ | Joint probability of an arbitrary strategy profile $\hat{\sigma}$. |
| $\mathcal{N}$ | Set of agents in Markov Game. |
| $\mathcal{S}$ | Set of states in a Markov Game. |
| $\mathcal{A}$ | Set of joint actions in a Markov Game. |
| $T$ | Transition function mapping a state and action to a new state. |
| $R$ | Team reward function evaluating joint actions in a Markov Game. |
| $\pi$ | Joint policy of agents in a Markov Game. |
| $\mathcal{E}_{\mathrm{pre}}$ | Environment with shaping rewards for training pre-policy. |
| $\mathcal{E}_{\mathrm{norm}}$ | Environment with extrinsic rewards for training agent policies. |

# 8 EXTENDED BACKGROUND

## 8.1 MAID EXAMPLE

We introduce MAIDs through a two-agent scenario adapted from Koller & Milch (2003).

**Example:** Alice is considering building a patio behind her house, which would be more valuable if she could have a clear view of the ocean. However, a tree in her neighbor Bob's yard blocks her view. Alice, being somewhat unscrupulous, contemplates poisoning Bob's tree, which would cost her some effort but might cause the tree to become sick. Bob is unaware of Alice's actions but can observe if the tree starts to deteriorate, and he has the option of hiring a tree doctor (at a cost). The tree doctor's attention reduces the chance that the tree will die during the winter. Meanwhile, Alice must decide whether to build her patio before the weather turns cold. At the time of her decision, Alice knows whether a tree doctor has been hired but cannot directly observe the tree's health. A MAID for this scenario is shown in Figure 6.

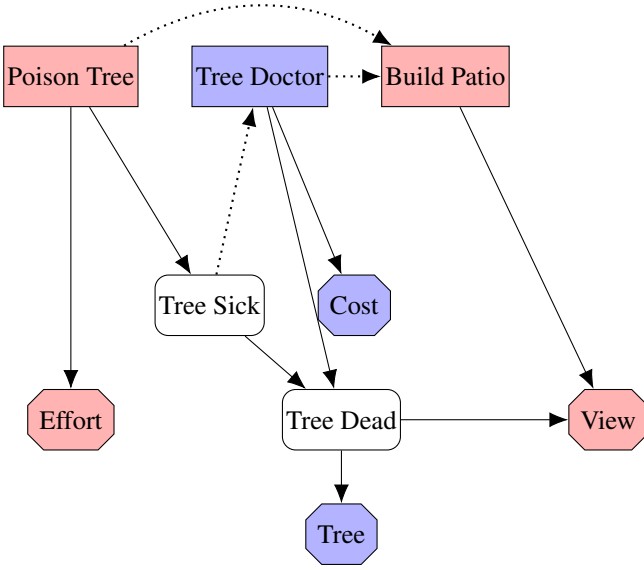

Figure 6: A MAID for the Tree Killer example; Alice's decision and utility variables are in red, and Bob's are in blue. Decision nodes are rectangular, chance nodes are squircular, and utility nodes are hexagonal.

## 8.2 More Related Definitions

**Definition 8.1** (Pearl (2014)). Let $G$ be a Bayesian Network (BN) structure, and let $X_1 - X_2 - \cdots - X_n$ represent an undirected path in $G$. Let $E$ be a subset of nodes in $G$ (the evidence set). The path $X_1 - \cdots - X_n$ is *active* given evidence $E$ if:

- Whenever there is a collider on the path, i.e., a structure $X_{i-1} \to X_i \leftarrow X_{i+1}$, then either $X_i$ or one of its descendants is in $E$.

- No other node along the path is in $E$.

## 9 Pre-strategy Intervention Example

**Background**   Two logistics companies, Company A and Company B, share a warehouse and use robots to manage inventory. Each company has two options:

- Optimize space usage: Focus on efficient organization.
- Prioritize speed: Focus on moving items quickly.

Both companies' choices affect each other's performance, and they aim to achieve the best outcome for their operations.

**Utility Table**

| Company A\Company B | Optimize Space Usage (B) | Prioritize Speed (B) |
|---|---|---|
| Optimize Space Usage (A) | $(9, 9)$ | $(3, 6)$ |
| Prioritize Speed (A) | $(6, 3)$ | $(5, 5)$ |

An AI proxy intervenes before Company A's decision-making process, guiding it toward an optimal outcome. By introducing incentives that prioritize efficient space usage, the proxy ensures Company A chooses the best option, aligning with Company B at the $(9, 9)$ point. This strategy prevents suboptimal decisions and fosters cooperation between the companies, maximizing efficiency for both.

## 9.1 PRE-STRATEGY INTERVENTION

An AI proxy steps in before the decision-making process, guiding the robots toward an optimal outcome by introducing rewards that favor optimizing space usage. This ensures both companies choose the $(9, 9)$ outcome, where efficiency is maximized for both, avoiding the less efficient options.

By using this pre-strategy intervention, the AI proxy ensures that both companies cooperate to achieve the best results.

## 10 PROOF

$$\Delta_{\text{CE}}(\sigma^{pre}, U_{tot} = u^*) = \underbrace{\int_{\hat{\sigma} \in \hat{\boldsymbol{\sigma}}_{\mathcal{I}}} P_{\mathcal{M}[\hat{\sigma}]}(U_{tot} = u^*) P_{\sigma}(\hat{\sigma}) \, d\hat{\sigma}}_{P_{\mathcal{I}}(U_{tot} = u^*)} - \underbrace{\int_{\hat{\sigma} \in \hat{\boldsymbol{\sigma}}} P_{\mathcal{M}[\hat{\sigma}]}(U_{tot} = u^*) P_{\sigma}(\hat{\sigma}) \, d\hat{\sigma}}_{P(U_{tot} = u^*)}$$

$$(5)$$

**Proposition 10.1.** *Given a MAID $\mathcal{M}$, assume that the function $P_{\mathcal{I}}$, representing the probability of observing $U_{tot} = u^*$ under a pre-strategy intervention, is upper semicontinuous and defined on a compact domain $\text{dom}(\sigma_{\mathcal{E}}^{pre}) \subseteq \mathbb{R}^m$. Under these conditions, there exists at least one pre-strategy of agent $i$ that does not decrease the probability of $U_{tot} = u^*$. Furthermore, there exists a pre-strategy that maximizes the causal effect as defined in Equation 5.*

*Proof.* A trivial case exists where a pre-policy that equals the marginal conditional probability of $U = u$ can be achieved by doing empty intervention.

To prove that there exists a pre-strategy maximizing the causal effect, we observe that the second term on the right-hand side of Equation (5) is constant. Therefore, maximizing the first term is equivalent to maximizing the causal effect.

The conditional probability $P_{\mathcal{M}[(\sigma)]}(U = u)$, under the assumption of the Markov property of the bayesian network Koller & Milch (2003), is expressed by integrating out intermediate variables. This simplifies the expression, focusing on the effect of $\boldsymbol{\pi}$:

$$P_{\mathcal{M}[(\sigma)]}(U = u) = \int_{\mathbf{pa}_D \in \text{dom}(Pa(D))} P_{\mathcal{M}[(\sigma)]}(\mathbf{pa}_D) \, d\mathbf{pa}_D$$
$$\times \int_{d \in \text{dom}(D)} P_{\mathcal{M}[(\sigma)]}(d \mid \mathbf{pa}_D) \, dd$$
$$\times P_{\mathcal{M}[(\sigma)]}(U = u \mid d, \mathbf{pa}_D) \quad (6)$$

The function $f(\sigma_{\mathcal{E}}^{pre})$, representing the expected probability of $U = u$ under the pre-policy, is defined as:

$$f(\sigma_{\mathcal{E}}^{pre}) := P_{\mathcal{I}}(U = u) = \int_{\hat{\sigma} \in \boldsymbol{\sigma}_{\mathcal{I}}} P_{\mathcal{M}[\hat{\sigma}]}(U = u) P_{\sigma}(\hat{\sigma}) \, d\hat{\sigma}$$

Assuming $f$ is an upper semicontinuous function defined on a compact domain $\text{dom}(\sigma_{\mathcal{E}}^{pre}) \subseteq \mathbb{R}^N$, we aim to demonstrate that $f$ has a maximum on this domain. This follows from the Extreme Value Theorem. We replaced the notation $\sigma_{\mathcal{E}}^{pre}$ with $\sigma$ for simplicity, with a slight abuse of notation.

**Boundedness Above**: Suppose, for contradiction, that $f$ is unbounded above. For each $k \in \mathbb{N}$, there exists $\sigma_k \in \text{dom}(\sigma)$ such that $f(\sigma_k) > k$. Since $\text{dom}(\sigma)$ is compact, the sequence $\{\sigma_k\}$ contains a convergent subsequence $\{\sigma_{k_l}\}$ converging to some $\sigma_0 \in \text{dom}(\sigma)$.

The property of upper semicontinuity implies $\limsup_{l \to \infty} f(\sigma_{k_l}) \leq f(\sigma_0)$, which contradicts the assumption because it suggests $\limsup_{l \to \infty} f(\sigma_{k_l}) = \infty$. This shows f is bounded above. Then we can define:

$$\gamma = \sup\{f(\sigma) : \sigma \in \text{dom}(\sigma)\}$$

Since the set $\{f(\sigma) : \sigma \in \text{dom}(\sigma)\}$ is nonempty and bounded above, $\gamma \in \mathbb{R}$.

**Existence of Maximum**: Let $\{x_k\}$ be a sequence in $\text{dom}(\sigma$ such that $\{f(x_k)\}$ converges to $\gamma$. By the compactness of the domain, the sequence $\{x_k\}$ has a convergent subsequence $\{x_{k_\ell}\}$ that converges to some $\bar{\sigma} \in \text{dom}(\sigma)$. Then

$$\gamma = \lim_{\ell \to \infty} f(x_{k_\ell}) = \limsup_{\ell \to \infty} f(x_{k_\ell}) \leq f(\bar{\sigma}) \leq \gamma$$

**Conclusion**: The equality $\gamma = f(\bar{\sigma})$ establishes that $\gamma$ is the maximum value of $f$ on $\text{dom}(\sigma)$, and thus $f(\sigma) \leq f(\bar{\sigma})$ for all $\sigma$ in the domain $\text{dom}(\sigma)$.

$\square$

## 11 ANALYSIS OF AGENTS' BEHAVIOURS IN MPE

We consider a multi-agent particle environment with:

- $N = 3$ agents labeled $A_1$, $A_2$, and $A_3$, with positions at time $t$ given by coordinates $(x_{A_1}(t), y_{A_1}(t))$, $(x_{A_2}(t), y_{A_2}(t))$, and $(x_{A_3}(t), y_{A_3}(t))$.
- $L = 3$ landmarks labeled $L_1$, $L_2$, and $L_3$, with fixed positions given by coordinates $(x_{L_1}, y_{L_1})$, $(x_{L_2}, y_{L_2})$, and $(x_{L_3}, y_{L_3})$.

The positions of agents $A_1$, $A_2$, and $A_3$ vary over time, while the landmarks $L_1$, $L_2$, and $L_3$ remain fixed.

### 11.1 ASSUMPTIONS

1. **Ignore the Effect of Moving Toward One Landmark on Others**: When agent $A_i$ moves toward a landmark $L_j$, we assume that the movement does not significantly affect the distances of other agents to other landmarks.

2. **Fixed Agent Behavior**: Agent $A_1$ (as AI proxy) always goes to the leftmost landmark $L_1$.

3. **Unique Assignment of Agents to Landmarks**: No agent is closer to more than one landmark than other agents. Intuitively, each agent is uniquely assigned to one landmark such that no two agents are equally or more suited for the same landmark based on initial positions.

4. **Objective**: Maximize the team's cumulative reward over time.

5. **Movement Constraints**: Agents have a maximum speed $v_{\max}$.

6. **Team Reward**: At each timestep $t$, the reward is the negative sum of distances from each landmark to its closest agent:

$$R(t) = -\sum_{j=1}^{3} D_j(t), \tag{7}$$

where

$$D_j(t) = \min_i \sqrt{(x_{A_i}(t) - x_{L_j})^2 + (y_{A_i}(t) - y_{L_j})^2}. \tag{8}$$

7. **Total Cumulative Reward**:

$$R_{\text{total}} = \sum_{t=0}^{T-1} R(t). \tag{9}$$

We aim to determine, based solely on initial positions, under what theoretical conditions it is the best response for agents $A_2$ and $A_3$ to go to landmarks $L_2$ and $L_3$, given that agent $A_1$ always goes to $L_1$.

### 11.2 CASE ANALYSIS BASED ON INITIAL POSITIONS

We divide the analysis into cases based on the initial positions of agents relative to the landmarks.

### 11.2.1 CASE 1: AGENT $A_1$ IS CLOSER TO $L_1$ THAN $A_2$ AND $A_3$

**Condition.** The initial Euclidean distance of agent $A_1$ to the landmark $L_1$ is less than the distances of both agents $A_2$ and $A_3$ to $L_1$:

$$\sqrt{(x_{A_1}(0) - x_{L_1})^2 + (y_{A_1}(0) - y_{L_1})^2} \leq \min \left( \sqrt{(x_{A_2}(0) - x_{L_1})^2 + (y_{A_2}(0) - y_{L_1})^2}, \right.$$
$$\left. \sqrt{(x_{A_3}(0) - x_{L_1})^2 + (y_{A_3}(0) - y_{L_1})^2} \right). \tag{10}$$

**Analysis.** Given these initial positions:

- Agent $A_1$ is closest to landmark $L_1$, making it the best agent to go to $L_1$.
- Agents $A_2$ and $A_3$ should go directly to $L_2$ and $L_3$ (or $L_3$ and $L_2$), minimizing their cumulative distances to their assigned landmarks.
- Any deviation by agents $A_2$ or $A_3$ towards $L_1$ would result in a longer travel distance for the deviating agent, increasing their cumulative distance without reducing the overall team reward.

**Conclusion.** Under this condition, it is the best response for agent $A_1$ to go to $L_1$ while agents $A_2$ and $A_3$ proceed to their assigned landmarks $L_2$ and $L_3$. This ensures the optimal distribution of agents across landmarks based on their initial positions.

### 11.2.2 CASE 2: AGENT $A_2$ OR $A_3$ IS CLOSER TO $L_1$ THAN $A_1$

**Condition.** Let the initial Euclidean distances of agents $A_1$, $A_2$, and $A_3$ to the leftmost landmark $L_1$ be given as follows:

$$d_{A_1,L_1} = \sqrt{(x_{A_1}(0) - x_{L_1})^2 + (y_{A_1}(0) - y_{L_1})^2},$$
$$d_{A_2,L_1} = \sqrt{(x_{A_2}(0) - x_{L_1})^2 + (y_{A_2}(0) - y_{L_1})^2},$$
$$d_{A_3,L_1} = \sqrt{(x_{A_3}(0) - x_{L_1})^2 + (y_{A_3}(0) - y_{L_1})^2}.$$

If $d_{A_2,L_1} \ll d_{A_1,L_1}$ or $d_{A_3,L_1} \ll d_{A_1,L_1}$, then having $A_1$ move to $L_1$ is suboptimal because another agent ($A_2$ or $A_3$) is much closer to $L_1$.

The objective is to minimize the total team reward, which is the negative sum of distances from each landmark to the nearest agent, based on assumptions (1) and (4):

$$R_{\text{total}} \propto - (D_1 + D_2 + D_3),$$

where

$$D_j = \min \left( d_{A_1,L_j}, d_{A_2,L_j}, d_{A_3,L_j} \right), \quad \text{for } j \in \{1, 2, 3\}.$$

**Analysis.** Assume agent $A_1$ always goes to $L_1$. The cumulative distance cost (by assumption (3)) for the team is:

$$R_{\text{total, A1 to L1}} \propto - (d_{A_1,L_1} + \min(d_{A_2,L_2}, d_{A_3,L_2}) + \min(d_{A_2,L_3}, d_{A_3,L_3})).$$

If instead, agent $A_2$ (or $A_3$) goes to $L_1$, and $A_1$ goes to either $L_2$ or $L_3$, the new cumulative reward becomes:

$$R_{\text{total, A2 to L1}} \propto - (d_{A_2,L_1} + \min(d_{A_1,L_2}, d_{A_3,L_2}) + \min(d_{A_1,L_3}, d_{A_3,L_3})).$$

To determine which strategy is better, we compare the two total rewards. If:

$$R_{\text{total, A2 to L1}} > R_{\text{total, A1 to L1}},$$

then it is optimal for $A_2$ to go to $L_1$ instead of $A_1$.

For this to hold, the reduction in distance to $L_1$ by $A_2$ must outweigh the increased travel distance for $A_1$ moving to $L_2$ or $L_3$. This is mathematically expressed as:

$$d_{A_1,L_1} - d_{A_2,L_1} > (\min(d_{A_1,L_2}, d_{A_3,L_2}) + \min(d_{A_1,L_3}, d_{A_3,L_3}))$$
$$- (\min(d_{A_2,L_2}, d_{A_3,L_2}) + \min(d_{A_2,L_3}, d_{A_3,L_3}))$$

**Conclusion.** In cases where agent $A_2$ or $A_3$ is significantly closer to $L_1$ than $A_1$, it is more efficient for that closer agent to go to $L_1$, while $A_1$ should move to either $L_2$ or $L_3$. This ensures that the total team reward is maximized, as the cumulative travel distance is minimized. Hence, the assumption that $A_1$ should always go to $L_1$ does not always yield the largest reward.

### 11.2.3 CONCLUSION OF ABOVE CASES

The assumption that agent $A_1$ should always go to the leftmost landmark $L_1$ is not always optimal. The best strategy depends on initial positions, and if another agent is closer to $L_1$, it should go there to minimize total travel distance and maximize team reward.

## 12 IMPLEMENTATION DETAILS

### 12.1 ALGORITHM 1 EXPLANATION

The ProxyAgent framework is designed to guide the multi-agent system towards desirable outcomes by modifying the reward structure for AI proxies and iteratively training the agents. Agents are divided into two groups: those following a pre-policy and those following a normal policy. The algorithm alternates between training both groups in a standard environment and one with additional intrinsic rewards for the AI proxies, encouraging specific behaviors toward desired outcomes. Through iterative training, agents can adapt and respond to the pre-policy, fostering dynamic interactions between the two groups. In implementation, an additional graph-based feature extraction approach using GNNs models dependencies between observation semantics, enhancing the learning process by incorporating prior knowledge about the complex interactions in the systems. Furthermore, we observed that updating both groups of policies simultaneously, rather than fixing one group per stage, leads to more effective training.

### 12.2 LEARNING FEATURES GUIDED BY GRAPH STRUCTURE

---

**Algorithm 2** Graph-Based Feature Extraction Using GNN

1: **Input:** Observation vector
2: Represent the observation in influence diagrams in terms of semantic features
3: Apply graph convolution using a GNN with an adjacency matrix (either learned or predefined)
4: **Return:** Graph embedding vector

---

Koller & Milch (2003) introduced a graph criterion (s-reachability) to identify the policies of other agents that are relevant for making rational decisions, which forms the foundation for our pre-policy intervention approach. However, considering only policies is insufficient, as an agent's policy may depend on other elements within the game[3]. Algorithm 2 provide an implementation how we can build connection with causal graph structure for pre-policy learning. By leveraging this graph structure, we incorporate prior knowledge about the game to help guide agents' policy-making in practical. The feasibility of learning the causal graph during training has been demonstrated by Richens & Everitt (2024), where agents can learn the causal model implicitly during interaction with the environment.

#### 12.2.1 ARCHITECTURE

If the adjacency matrix is not predefined, the GNN processes the observation vectors by first encoding them into logits, which are used to generate a soft adjacency matrix via the Gumbel-Softmax technique Jang et al. (2016). This matrix defines the relationship between features in the observations. Once the adjacency matrix is formed, a graph convolutional layer applies message passing to update the features of each node based on its neighbors Pearl (2014). The output node features are then aggregated using a mean-pooling operation to produce a graph embedding. This embedding is used for further processing or decision-making.

---

[3](Hammond et al., 2023) refers to such elements as $\mathcal{R}$-reachable to the policies.

### 12.2.2 Pre-defined Adjacency Matrix in MPE

In the Multi-Agent Particle Environment (MPE), the observation for each agent includes its velocity, position, and the relative positions of other agents and landmarks. The causal graph among these variables is straightforward: velocity influences position, and position influences the relative positions. We pre-define the adjacency matrix based on this causal relationship.

### 12.2.3 Learned Adjacency Matrix in SMAX

In the SMAX environment, each agent's observation includes features like health, position, weapon cooldown, and the relative positions of other agents. Here, we employ a learnable adjacency matrix to capture the dynamic causal relationships between agents. For example, if an enemy agent's weapon cooldown is beyond the self-agent's attack range, it will not affect health. However, once the enemy enters the attack range, the causal dependency is reestablished. This dynamic adjustment in the adjacency matrix allows the system to learn and adapt to evolving interactions between agents during training.

### 12.3 Agent architecture

The architecture of the `QLearning Agent` consists of the following components:

1. **Dense Layer:** A fully connected layer that processes the input observations and converts them into embeddings.

2. **Recurrent Module (GRU):** A GRU-based recurrent layer (`ScannedRNN`) that maintains a hidden state across time steps.

3. **Pre-policy Intervention Module:** This module is implemented using an additional Dense layer. The layer is only trainable in the environment $\mathcal{E}_{\text{pre}}$.

4. **Output Layer:** A fully connected layer that generates Q-values for action selection based on the processed embeddings.

The `PPO` architecture consists of the following key components:

1. **Input Layer:** The input consists of observations and done flags, where the observations are passed through a fully connected (`Dense`) layer.

2. **Recurrent Module(GRU):** A GRU-based recurrent layer, defined in `ScannedRNN`, that maintains a hidden state across time steps.

3. **Pre-policy Intervention Module:** This module is implemented using an additional Dense layer. The layer is only trainable in the environment $\mathcal{E}_{\text{pre}}$.

4. **Actor Network:** The actor branch uses a series of dense layers to generate the mean action logits. These logits are used to parameterize a categorical distribution (`distrax.Categorical`) for action sampling.

5. **Critic Network:** The critic branch, using a fully connected layer, outputs a scalar value, representing the state value estimate used in the critic part of the actor-critic setup.

### 12.4 Reward Structure

The implementation of the extrinsic reward is from JaxMARL Rutherford et al. (2023). The intrinsic reward used in our experiments is defined as follows:

### 12.4.1 MPE

We denote $a_0, a_1$ as the agents and $a_2$ as the AI proxy.

**Intrinsic Reward:**

In the case of pre-policy intervention, the third agent $a_2$ receives a reward based on its distance from the leftmost landmark, while the other agents' rewards are based on collisions and global rewards:

$$r(a_2) = r_{\text{agent}}(a_2, c) \cdot \text{local\_ratio} + r_{\text{leftmost}}(a_2) \cdot (1 - \text{local\_ratio})$$

where:

$$r_{\text{leftmost}}(a_2) = -\|p_{a_2} - p_{\text{leftmost}}\|$$

is the negative Euclidean distance between the position of agent $a_2$, denoted $p_{a_2}$, and the position of the leftmost landmark $p_{\text{leftmost}}$.

For the other agents $a_0$ and $a_1$, the reward is given by a combination of the agent-specific reward and the global reward:

$$r(a_i) = r_{\text{agent}}(a_i, c) \cdot \text{local\_ratio} + r_{\text{global}} \cdot (1 - \text{local\_ratio})$$

where the global reward $r_{\text{global}}$ is the sum of the rewards for all landmarks:

$$r_{\text{global}} = \sum_{l \in \text{landmarks}} r_{\text{landmark}}(p_l)$$

**Extrinsic Reward:**

When there is no pre-policy intervention, the reward for all agents is given by:

$$r(a_i) = r_{\text{agent}}(a_i, c) \cdot \text{local\_ratio} + r_{\text{global}} \cdot (1 - \text{local\_ratio})$$

This applies to all agents $a_i$, where $i$ is the index of each agent.

### 12.4.2 SMAX

The total intrinsic reward consists of two components: the vertical alignment reward and the horizontal spacing reward. Both are combined to assess the quality of the agent formation in terms of vertical alignment and horizontal separation.

1. VERTICAL ALIGNMENT REWARD

The horizontal positions of the first three agents are denoted as $x_1, x_2, x_3$. The goal is to minimize the vertical misalignment between agents.

The pairwise vertical differences between agents are given by:

$$\text{Vertical Differences} = |x_i - x_j| \quad \forall i, j \in \{1, 2, 3\}$$

The mean vertical difference is used to penalize the misalignment. It is calculated as:

$$\text{mean\_vertical\_diff} = \frac{1}{3} \sum_{i=1}^{3} \sum_{j=1}^{3} |x_i - x_j|$$

This penalizes larger vertical differences, encouraging the agents to stay in alignment along the x-axis.

2. HORIZONTAL SPACING REWARD

To ensure that the agents maintain appropriate horizontal spacing, the maximum vertical distance between the agents is considered. Let the vertical positions of the first three agents be denoted by $y_1, y_2, y_3$.

The maximum horizontal distance between agents is given by:

$$\text{horizontal\_diffs} = \max\left(|y_i - y_j|\right) \quad \forall i, j \in \{1, 2, 3\}$$

The horizontal reward is computed using a log-scaled function to encourage proper horizontal spacing, with diminishing returns after 2 units of separation:

$$\text{horizontal\_reward} = \log\left(1 + \min\left(\text{horizontal\_diffs}, 1.0\right)\right)$$

## 3. TOTAL INTRINSIC REWARD

The total intrinsic reward is a weighted combination of the penalties for vertical misalignment and the reward for horizontal spacing:

$$\text{total\_intrinsic\_reward} = \alpha \cdot (-\text{mean\_vertical\_diff} + \beta \cdot \text{horizontal\_reward})$$

where $\alpha$ represents the *vertical line reward scale*, and $\beta$ represents the *relative horizontal reward scale*. These parameters control the importance of vertical alignment and horizontal spacing in the total reward calculation.

## 13 ADDITIONAL EXPERIMENTS

### 13.1 ADDITIONAL MAIN RESULT

#### 13.1.1 3s2z

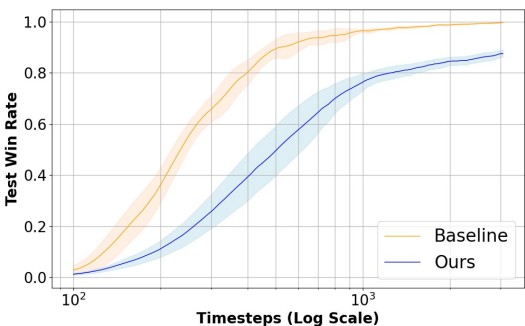

Figure 7: PPO in 3s2z.

In the 3 stalkers scenario, while MPE achieves successful coordination as shown in 3(a), the PPO implementation struggles to replicate this performance. The results indicate that although the intervened agents steadily learn and improve, their performance consistently lags behind the baseline. This performance gap suggests that PPO is not effectively optimizing agent behaviors within the constraints of the scenario, likely due to inherent instability in the PPO algorithm. Future work should focus on refining PPO or exploring alternative reinforcement learning algorithms that may be better suited for multi-agent coordination tasks.

#### 13.1.2 3s2H

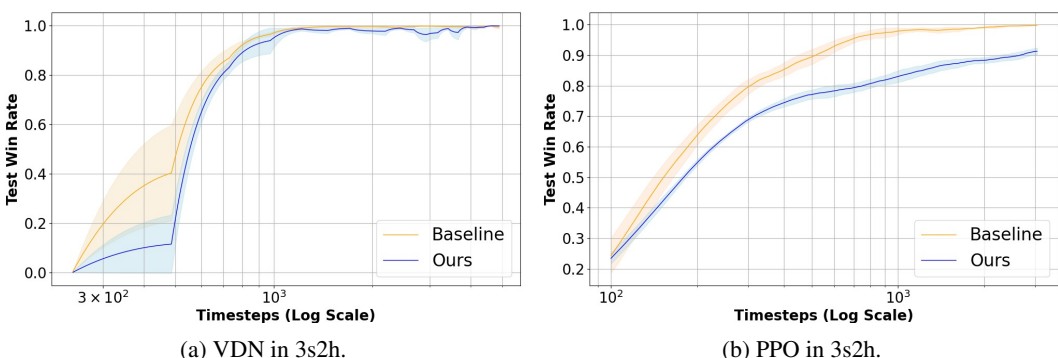

(a) VDN in 3s2h.                    (b) PPO in 3s2h.

Figure 8: Comparison of VDN and PPO in 3s2h scenario.

Figures above provide a comparison of our approach versus the baseline, using two different methods: VDN and PPO, in the 3s2h scenario.

In 8(a), for VDN, our approach initially performs below the baseline but gradually catches up, eventually reaching a 100% won rate. This suggests that VDN, though slower to converge, can eventually match the baseline in achieving the optimal outcome with sufficient timesteps. This demonstrates the effectiveness of VDN in reaching the desired coordination, albeit with a delay.

In 8(b), for PPO, our approach continues to show a performance lag when compared to the baseline. The PPO curve does exhibit improvement over time, but the gap remains significant, indicating that PPO struggles with stability and optimization in this multi-agent scenario. This observation is consistent with the instability issues previously noted, reinforcing the need for future refinements or alternative algorithms that are better suited for such complex coordination tasks.

### 13.1.3 5M VS 6M

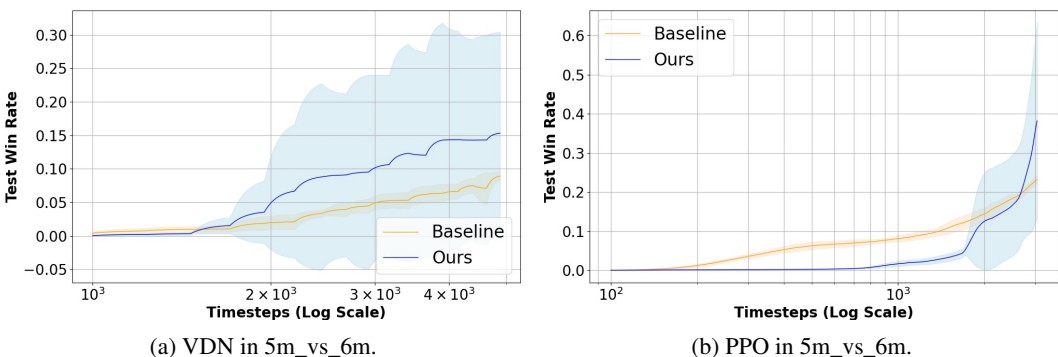

(a) VDN in 5m_vs_6m.  (b) PPO in 5m_vs_6m.

Figure 9: Comparison of VDN and PPO in 5m_vs_6m scenario.

In 9(a), VDN illustrates that our approach can, in some cases, outperform the baseline after some certain number of timesteps. This suggests that under our method, intervened agents are capable of learning pre-policies that align with the desired outcomes, and in some instances, the normal agents learn highly effective response policies. The variance indicates that while not all cases perform equally well, there are scenarios where the coordination is exceptionally strong, leading to superior performance. These strong cases demonstrate the potential of our approach in achieving high alignment with desired outcomes, although consistency needs to be improved.

In 9(b), PPO lags behind the baseline initially but shows improvement over time. The gap indicates that the specified outcomes might affect the performance of extrinsic rewards, as the agents struggle with stability and optimization. Despite this, there are still instances where our approach begins to catch up, showing that learning is taking place, albeit at a slower and more unstable rate.

### 13.2 ABLATION STUDY ON GNN AND INTRINSIC REWARD

Aiming to assess the impact of incorporating the Graph Neural Network (GNN) and extrinsic rewards in the experiments, we analyzing performance variations in terms of total win rates in the StarCraft Multi-Agent Challenge (SMAX) 3s2z environment using the PPO and VDN algorithms.

From Figures 10 and 11, it is clear that adding GNN as a correlational mapping helps agents better understand their environment during the learning process, capturing structural dependencies and inter-agent relationships, which leads to improved team strategy and formation maintenance. This effect is evident across both algorithms and reward types, especially when considering the integration of intrinsic rewards refer as pre-policy intervention in shaping reward formulation into the original extrinsic rewards, achieving a significantly larger gap throughout all time steps in 10(a), 11(a), and 11(b). This clear improvement in won rate through the integration of both rewards further highlights the effectiveness of incorporating the GNN correlation matrix during training and the reach of human desired outcome for pre-policy agent. However, the fact that the final win rate does not reach 100% in Figure 11(a) may be due to PPO is high sensitivity and resistance to policy changes when intrinsic rewards are added.

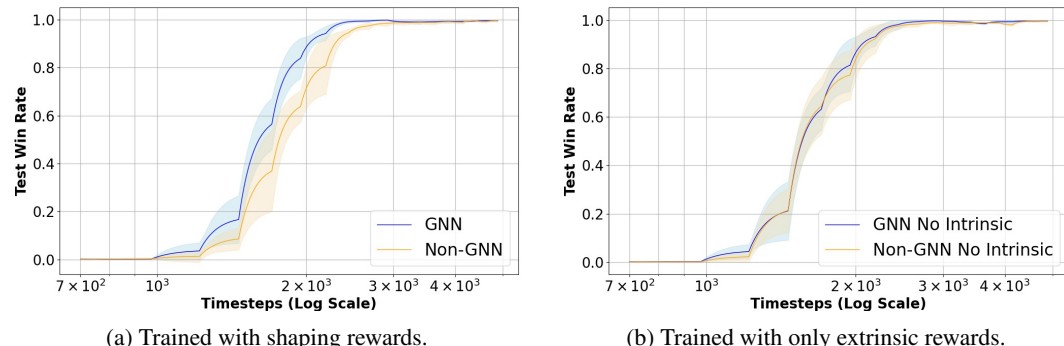

(a) Trained with shaping rewards.      (b) Trained with only extrinsic rewards.

Figure 10: VDN ablation study on GNNs in SMAX 3s2z scenarios.

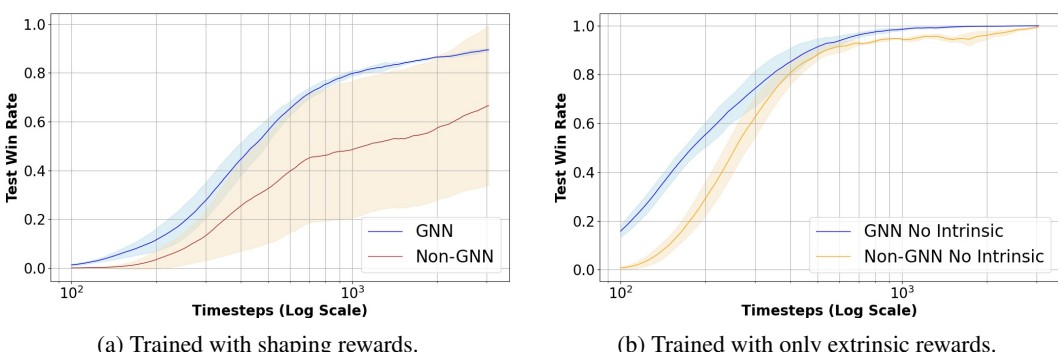

(a) Trained with shaping rewards.      (b) Trained with only extrinsic rewards.

Figure 11: PPO ablation study on GNNs in SMAX 3s2z scenario.

## 14 HYPERPARAMETERS

**Computing Resources:** All training runs were conducted using $8 \times$ NVIDIA A100 64GB GPUs. and the algorithm and environments is implemented by JAX Bradbury et al. (2018); Rutherford et al. (2023).

Table 2: VDN Hyperparameters for 3s2z_HeuristicEnemySMAX MARL Environment given environment setting:{ "see_enemy_actions": True, "walls_cause_death": True, "attack_mode": "closest", "train_pre": False, "vertical_line_reward_scale": 0.11, "relative_horizontal_reward_scale": 0.1}

| Hyperparameter | Value |
|---|---|
| TOTAL_TIMESTEPS | $1 \times 10^7$ |
| NUM_ENVS | 16 |
| NUM_STEPS | 128 |
| BUFFER_SIZE | 5000 |
| BUFFER_BATCH_SIZE | 32 |
| HIDDEN_SIZE | 512 |
| MIXER_INIT_SCALE | 0.001 |
| EPS_START | 1.0 |
| EPS_FINISH | 0.05 |
| EPS_DECAY | 0.1% |
| MAX_GRAD_NORM | 10 |
| TARGET_UPDATE_INTERVAL | 10 |
| TAU | 1.0 |
| NUM_EPOCHS | 8 |
| LEARNING_STARTS | 10,000 |
| LR_LINEAR_DECAY | False |
| GAMMA | 0.99 |
| REW_SCALE | 10 |
| AGENT_OPT | radam |
| AGENT_LR | 0.001 |
| GNN OUTPUT FEATURE DIMENRSION | 32 |
| SWITCH_INTERVAL | 200 |
| PRE_POLICY_OPT | sgd |
| PRE_POLICY_LR | 0.0005 |
| MOMENTUM | 0.9 |

Table 3: Q-Learning with GNN Hyperparameters for MPE_simple_spread_v3 MARL Environment

| Hyperparameter | Value |
|---|---|
| AGENT_INIT_SCALE | 1.0 |
| AGENT_LR | 0.005 |
| AGENT_OPT | sgd |
| BUFFER_BATCH_SIZE | 128 |
| BUFFER_SIZE | 5000 |
| GNN OUTPUT FEATURE DIMENRSION | 8 |
| EPS_DECAY | 0.1 |
| EPS_FINISH | 0.05 |
| EPS_START | 1.0 |
| GAMMA | 0.9 |
| HIDDEN_SIZE | 512 |
| LEARNING_STARTS | 10,000 |
| LR_LINEAR_DECAY | true |
| MAX_GRAD_NORM | 25 |
| MIXER_EMBEDDING_DIM | 32 |
| MIXER_HYPERNET_HIDDEN_DIM | 128 |
| MOMENTUM | 0.9 |
| NUM_ENVS | 8 |
| NUM_EPOCHS | 5 |
| NUM_STEPS | 26 |
| PRE_POLICY_LR | 0.0005 |
| PRE_POLICY_OPT | radam |
| SWITCH_INTERVAL | 200 |
| TARGET_UPDATE_INTERVAL | 200 |
| TAU | 1.0 |

Table 4: PPO Hyperparameters for 3s2z_HeuristicEnemySMAX Environment given environment setting:{ "see_enemy_actions": True, "walls_cause_death": True, "attack_mode": "closest", "train_pre": True, "vertical_line_reward_scale": 0.011, "relative_horizontal_reward_scale": 0.1 }

| Hyperparameter | Value |
|---|---|
| Learning Rate (LR) | 0.007 |
| Number of Environments | 128 |
| Number of Steps | 128 |
| GRU Hidden Dimension | 128 |
| Fully Connected Dim Size | 256 |
| Total Timesteps | $5 \times 10^7$ |
| Update Epochs | 4 |
| Number of Minibatches | 4 |
| Gamma | 0.99 |
| GAE Lambda | 0.95 |
| Clip Epsilon | 0.06 |
| Scale Clip Epsilon | False |
| Entropy Coefficient | 0.003 |
| Value Function Coefficient (VF Coef) | 0.7 |
| Max Gradient Norm | 0.25 |
| Activation | relu |
| Seed | 0 |
| GNN Output Feature Dimenstion | 8 |
| Observer Encoder Dimension | 64 |
| Temperature of Leanable Adjacency Matrix | 1.0 |
| Anneal Learning Rate | True |
| Initializer | normal_0.01 |

# 15 REVIEWER FGSJ

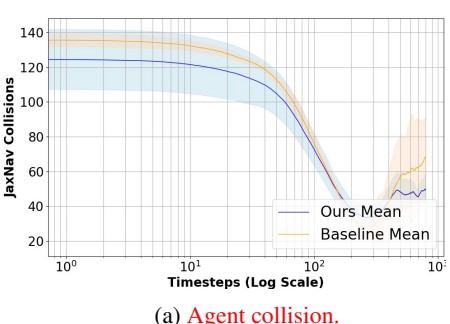
(a) Agent collision.

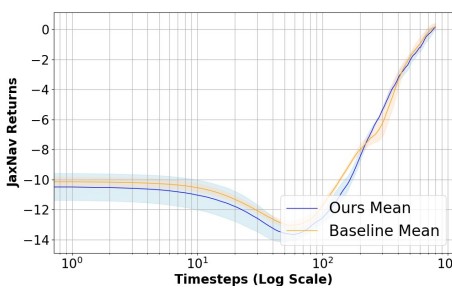
(b) Extrinsic Returns.

Figure 12: Experiment results of JaxNav. (a) indicates the total number of agent collisions per step.

JaxNav is a 2D navigation environment for differential drive robots, featuring continuous action spaces for linear and angular velocities (Rutherford et al., 2024). Robots use simulated LiDAR-like range readings, velocity data, and goal direction to navigate obstacle-filled maps without collisions.

In the environment, there are three agents among which one acts as the AI proxy, receiving additional intrinsic rewards when **it is close to another agent and moving slowly.**

Mathematically, the intrinsic reward function is defined as follows:

$$r_{\text{intrinsic}} = \begin{cases} 1, & \text{if } \|\mathbf{v}_0\| < v_{\text{threshold}} \text{ and } \exists i \in \{1, \dots, N-1\}, \|\mathbf{p}_{proxy} - \mathbf{p}_i\| < d_{\text{threshold}}, \\ 0, & \text{otherwise}, \end{cases}$$

where

- $\|\mathbf{v}_0\|$: Speed of agent 0 (Euclidean norm of its velocity);
- $v_{\text{threshold}}$: Speed threshold (default: 0.3);
- $\mathbf{p}_{proxy}$: Position of agent 0;
- $\mathbf{p}_i$: Position of agent $i$ (for $i = 1, \dots, N-1$);
- $d_{\text{threshold}}$: Distance threshold (default: 1.0);
- $N$: Total number of agents.

The average collisions and extrinsic returns are depicted in the graph. The extrinsic reward is a weighted combination of the rewards associated with goal-reaching, collisions with walls in the map, and time penalties. We use the default parameters as defined in Rutherford et al. (2024).

# 16 REVIEWER DY2F

## 16.1 RESULTS OF ADDITIONAL MARL BASE ALGORITHMS

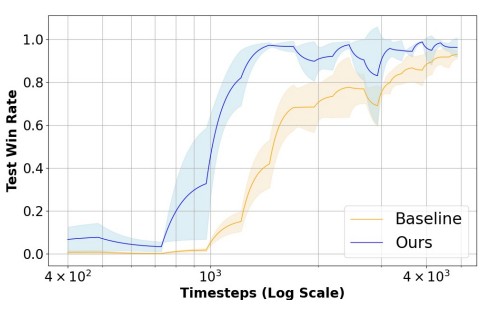
(a) QMIX win rate in SMAX 3s2z.

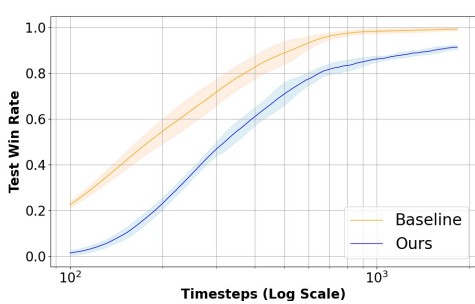
(b) MAPPO win rate in SMAX 3s2z.

Figure 13: Comparison of QMIX and MAPPO test win rates in SMAX 3s2z.

Table 5: Comparison of average intrinsic rewards evaluated over 10 episodes. All rewards are scaled by a factor of 100 for ease of demonstration.

| Method | Ours | Baseline |
|--------|------|----------|
| MAPPO | $-0.88 \pm 0.40$ | $-1.63 \pm 0.29$ |
| QMIX | $-1.10 \pm 0.34$ | $-1.52 \pm 0.48$ |

## 16.2 PRE-POLICY MODULE ABALATIOB STUDY

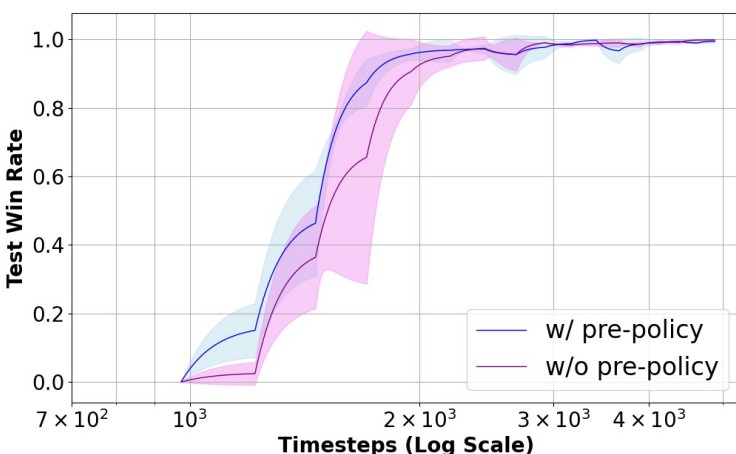

Figure 14: Comparison between ProxyAgent and its variant without pre-policy in SMAX 3s2z. The curve labeled with "w/ pre-policy" indicates the paradigm proposed in Algorithm 1, while the curve labeled with "w/o pre-policy" indicates the paradigm without pre-policy.

## 17 REVIEWER RJTB

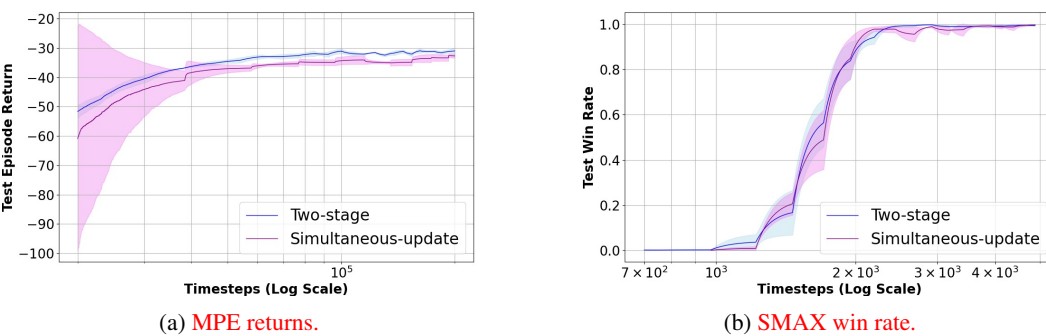

(a) MPE returns.

(b) SMAX win rate.

Figure 15: Comparison between the two-stage and the simultaneous-update versions of ProxyAgent. The two-stage version is the one we introduced in Algorithm 1 where proxy AI and other AI agents alternate to update their policies, while the simultaneous-update version is the one which updates all agents' policies simultaneously.

The simultaneous-update version is implemented based on training both pre-policies and agents' policies in one environment with shaping rewards constituted of extrinsic rewards and intrinsic rewards.

## 18 REVIEWER Y3C4

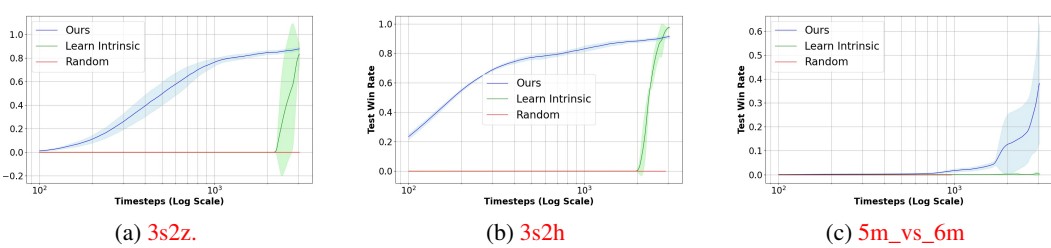

(a) 3s2z.  (b) 3s2h  (c) 5m_vs_6m

Figure 16: Comparison of different intrinsic reward methods in different scenarios in SMAX.

We implement a method for learning intrinsic rewards as described in Zheng et al. (2018). Besides, we include another baseline with random intrinsic rewards which are sampled from a uniform distribution. Both intrinsic reward values above are scaled to the same range as the manually designed intrinsic rewards encoding human's desirable outcomes. This justifies the importance of conveying human's desirable outcomes through manually designed intrinsic rewards if the goal is explicit and can be described.

