# OpenReview forum: "Attaining Human's Desirable Outcomes in Indirect Human-AI Interaction via Multi-Agent Influence Diagrams"
_ICLR.cc/2025/Conference — Submitted to ICLR 2025_

### Official Review · Reviewer_rjtB · 2024-11-04

**Soundness:** 2
**Presentation:** 2
**Contribution:** 2
**Rating:** 5
**Confidence:** 3

**Summary:**

This paper considers an _indirect_ human AI coordination problem where the humans influence a group AI agents by setting pre-strategy for certain specific agent or agents. It formulates the problem via multi-agent influence diagrams (MAIDs) and establish connections between MAIDs and MARL.

Disclaimer: I have to admit that I am not able to fully understand the theoretical aspect of this paper, so please take my opinion with a grain of salt.

**Strengths:**

* The problem setting, indirect human-AI coordination, seems novel. The paper establishes connections between this setting with MAIDs and makes some theoretical contributions.
* The connections can be used to provide alternative views for some common MARL challenges, such as the non-stationary issue of independent Q learning (IQL).

**Weaknesses:**

The problem setting of indirect human AI coordination is insufficiently motivated. It seems that the main problem it addresses is to make sure that the learned equilibrium align well with human intention. Such problems have been studied in the literature in much more complicated settings. For example, the Cicero agent for Diplomacy[1] is trained to converge to human-like equilibrium with QRE-inspired regret matching procedure  so that it can influence other players in a human-desired way. Hu & Sadigh [2] uses natural language to construct prior for one or multiple agents, which can also be seen as some form of shaped reward, so that they behave in a human-desired way when coordinating with other agents or other humans. I am not sure if this work is fundamentally different from those “equilibrium selection” work. In the experimental results, it seems that the paper is doing similar things, such as using extra reward shaping for certain agent so that the group exhibits certain collective behavior.

I think the paper could do a better job if it can
* Make a stronger motivation for indirection human AI coordination
* Clearly contrast it with existing works and emphasize on the unique advantages that prior work cannot achieve.

The related work section on LLM alignment seems less relevant for this paper as the author makes no comparison or application into the LLM land. On the contrary, it seems to miss some of the potentially related work mentioned above. However, it could also be that I missed the main point of the paper.

[1] Human-level play in the game of Diplomacy by combining language models with strategic reasoning

[2] Language Instructed Reinforcement Learning for Human-AI Coordination

**Questions:**

* Figure3: “Figure 3 shown that our method demonstrates general faster convergence compared with the baseline.” In Figure 3.a the GNN curve looks indistinguishable from the baseline curve, and in Figure 3.b and yellow baseline curve seems to converge faster. How do you reach the conclusion?

* MPE experiments: “the greater the distance the larger intrinsic rewards” -> Shouldn’t this drive the red dot further from the leftmost landmark? In Figure 4.a it seems that the red dot is moving closer to the leftmost landmark?

* Instead of using the two staged approach, how would the proposed method compare to a baseline that jointly optimize all agents while the proxy agent receives the intrinsic reward?

---

> ### Author Response · Authors · 2024-11-20
>
> We would like to thank reviewer rjtB's feedback on our novelty, theoretical contributions as well as our idea can also useful for some general tasks. We respond to each of the reviewer's questions in the sections below.
>
> ### Weaknesses
> > 1. Make a stronger motivation for indirection human AI coordination
>
> **Answer**: We agree with your discussion above. Indirect human-AI coordination can be treated as a kind of equilibrium selection problem from the layer of mathematical problem settings. From the perspective of application, we can provide another exmaple. Suppose a future scenario where a person named Bob need to interact with other AI agents (NPCs) and other AI proxies (can also be seen as AI agents to Bob in this case). Bob may not intend to expose himself to others because he would hide some private information (e.g. his facial profile). To satify this purpose, he has to hire an AI proxy.
>
> > 2. Clearly contrast it with existing works and emphasize on the unique advantages that prior work cannot achieve.
>
> The related work section on LLM alignment seems less relevant for this paper as the author makes no comparison or application into the LLM land. On the contrary, it seems to miss some of the potentially related work mentioned above. However, it could also be that I missed the main point of the paper.
>
> **Answer**: [2] actually solved the similar problem setting as we proposed (indirect human-AI interaction), though they may allow AI proxies to interact with other agents not limited to AI agents. On contrary, [1] actually implemented an agent that is able to infer human's intentions in direct human-AI interaction tasks. This example could be an evidence to justify the necessity of formalizing the difference between these two paradigms. Although both paradigms share some common solutions (e.g. the case you mentioned above), it does not imply that we have to use a more complicated method when we have some exclusive benefits of the problem setting. For example, in [2] human can directly give instructions to an AI assistant, while in [1] this is not allowed. On the other hand, inspired by the proposition in [2], we suppose that indirect human-AI could be a precondition to fulfil more complicated scenarios, such as AI proxies interacting with other humans and AI proxies interacting with other AI proxies on behalf of other humans. This could be another reason to formalize this problem setting.
>
> Second, we agree that [2] proposed to use intrinsic rewards to encode human's desirable outcomes to achieve human-desired equilibrium. However, it just treated intrinsic rewards as an approach without a justification of its validity at theoretical level. In our work, we propose to employ MAIDs, a graphical mdoel to justify the correctness in theory, which is the fundamental difference from these two previous works and a complementary work to support the reasonableness of [2]. Thanks for your so critical point. We have added this discussion in the related work. As for related work about LLM alignment, we agree with you and Reviewer Dy2f that it is with weak connection to our work. We have removed it in the revised paper.
>
> [1] Human-level play in the game of Diplomacy by combining language models with strategic reasoning
>
> [2] Language Instructed Reinforcement Learning for Human-AI Coordination

---

> ### Author Response · Authors · 2024-11-20
>
> ### Questions
>
> > 1. Figure3: “Figure 3 shown that our method demonstrates general faster convergence compared with the baseline.” In Figure 3.a the GNN curve looks indistinguishable from the baseline curve, and in Figure 3.b and yellow baseline curve seems to converge faster. How do you reach the conclusion?
>
> **Answer**: Thanks for your suggestion. We agree that our statement is not clear enough. First, in Figure 3a as the the combination of intrinsic rewards and extrinsic rewards may induce a new solution, our method can reach this new optimal solution (a NE that complies with human's desirable outcomes) faster at about $8\times10^{4}$ timesteps. However, with no influence by intrinsic rewards, the only optimal solution of baseline algorithm is maximizing the extrinsic rewards, which has not reached at least before the end of the demonstrating time axis. To justify the existence of effects of human's desirable outcomes, we also provide the average last-step intrinsic reward evaluation for MPE in Figure 4c in the revised paper. In Figure 3b, although the yellow baseline climbs steeply in the first half of training time, our method (in green) surpasses it afterwards.
>
> > 2. MPE experiments: “the greater the distance the larger intrinsic rewards” -> Shouldn’t this drive the red dot further from the leftmost landmark? In Figure 4.a it seems that the red dot is moving closer to the leftmost landmark?
>
> **Answer**: Thanks for your careful inspection. This is a typo. The correct version should be "the smaller the distance the larger intrinsic rewards." We have corrected it in the revised paper.
>
> > 3. Instead of using the two staged approach, how would the proposed method compare to a baseline that jointly optimize all agents while the proxy agent receives the intrinsic reward?
>
> **Answer**: Thanks for your insightful suggestion. We have given the results of this experiment in Appendix 17 in the revised paper. In summary, simultaneous-update approach would be similar to or a little weaker than the two-stage approach. This is not surprising, as simultaneous update can still optimize Equation (4). However, this is not an evidence to weaken the meaningfulness of the two-staged approach, as the disentanglement of these two optimization stages can open up the possibility of pretraining pre-policy. One can suppose that the time interval between these two stages is long enough, and the pre-policy training stage can be accomplished separately (pretrained) by the collected data from the provided human feedback as intrinsic rewards.

---

> > ### Comment · Reviewer_rjtB · 2024-11-27
> >
> > Thank you for your response and I appreciate the new experiment with single-stage reward shaping. I would maintain my rating as I found this paper a bit weak on the experimentation side. The authors have tried to polish the motivation to justify the theoretical necessity but it's a bit hard to grasp that with the current set of experiments.

---

> > > ### Author Response · Authors · 2024-11-27
> > >
> > > Thanks for your response. We believe in common rules if we have added the experiemnts you require, it may be worth considering to increase the rating, especially when the rating is 5. Although we agree that our experiments looks not as fancy as other expirical works, this cannot conceal the contribution of our work.
> > >
> > > To reach the final goal of the problem we proposed in this paper, it needs a long journey like many other research problems. In this paper, we only show an initial solution to the problem, which we believe is sufficient to show justify the possibility and difficulty of our proposed problem.
> > >
> > > We would still appreciate your generous time spent on our paper. However, we still hope you could reconsider your rating, with our further explanation above.

---

### Official Review · Reviewer_y3C4 · 2024-11-04

**Soundness:** 3
**Presentation:** 3
**Contribution:** 3
**Rating:** 6
**Confidence:** 3

**Summary:**

This paper studies a human-ai collaboration where the human interacts with other AI agents through an AI proxy. They model the indirect, interactive process among AI proxy and other agents using a multi-agent influence diagram and propose a pre-strategy intervention to select a human-desired NE. They use causal effects to optimize and measure the pre-strategy intervention in MARL. They study the MAID view of team reward in independent learning and centralized learning scenarios and evaluate proxy agents in MPE and StarCraft challenges to find (1) the proposed method converges faster than baselines and (2) human desirable outcomes that could affect the extrinsic rewards.

**Strengths:**

In general, it is a strong and well-written paper studying selecting the human-preferred Nash equilibrium and studying whether the algorithm works.

**Weaknesses:**

While the story and theory of the paper are interesting, they are less attractive in terms of method and experiments.

(1) In method/algorithm 1, I could barely get information besides the alternate training of pi_pre using intrinsic rewards and pi_agent with extrinsic rewards

(2) In experiment session, I would also demonstrate the method some games where the diverse NE(conventions) are easy to tell and has abundant prior work because nothing prevents the methods to be applied to the general-sum setting.

(3) Possibly missing literature on multi-agent diversity. I believe the diverse convention finding literature is relevant because you can find them in the first place and select among NEs. Here is a few examples [1][2][3]

[1] Hu H, Sadigh D. Language instructed reinforcement learning for human-ai coordination[C]//International Conference on Machine Learning. PMLR, 2023: 13584-13598.

[2] Cui, B., Lupu, A., Sokota, S., Hu, H., Wu, D. J., & Foerster, J. N. (2023). Adversarial diversity in Hanabi. In *The Eleventh International Conference on Learning Representations*.

[3] Rahman, M., Cui, J., & Stone, P. (2024, March). Minimum coverage sets for training robust ad hoc teamwork agents. In *Proceedings of the AAAI Conference on Artificial Intelligence* (Vol. 38, No. 16, pp. 17523-17530).

**Questions:**

(1) This paper reminds me of [1]. Can the authors elaborate on how language conditioning and pre-strategy are related to each other?

[1] Hu H, Sadigh D. Language instructed reinforcement learning for human-ai coordination[C]//International Conference on Machine Learning. PMLR, 2023: 13584-13598.

(2) How do you get the intrinsic rewards? How do humans specify their preferences?

---

> ### Author Response · Authors · 2024-11-20
>
> We thank reviewer y3C4 for their suggestions and appreciate that our paper is strong and well-written. We will address each of the questions posed by the reviewer in the sections below.
>
> ### Weaknesses
>
> While the story and theory of the paper are interesting, they are less attractive in terms of method and experiments.
>
> > 1. In method/algorithm 1, I could barely get information besides the alternate training of pi_pre using intrinsic rewards and pi_agent with extrinsic rewards
>
> **Answer**: We agree that the algorithm does not look fancy. However, we would like to emphasize that the principle of deriving the algorithm from our theory is a more critical contribution as you observed. In other words, the main purpose of the algorithm tends to verify the validity of our theory. Furthermore, there also exist other ways to implement our theory.
>
> > 2. In experiment session, I would also demonstrate the method some games where the diverse NE(conventions) are easy to tell and has abundant prior work because nothing prevents the methods to be applied to the general-sum setting.
>
> **Answer**: We agree that a case that can easily demonstrate the results may make people better understand our work. In our MPE experiment, we actually have provided a theoretical analysis of the result in Appendix 11, where diverse NEs are easy to tell.
>
> > 3. Possibly missing literature on multi-agent diversity. I believe the diverse convention finding literature is relevant because you can find them in the first place and select among NEs. Here is a few examples [1][2][3]
>
> **Answer**: Thank you for your suggestion. We understand your viewpoint. Although diverse multi-agent behaviours are preferred in our work, how these diverse behaviours can be generated are out of our research scope. For this reason, we cannot agree with your viewpoint at this moment. Nevertheless, [1] is related to human-ai coordination which we have added to our related work and have a discussion in the revised paper.
>
>
> [1] Hu H, Sadigh D. Language instructed reinforcement learning for human-ai coordination[C]//International Conference on Machine Learning. PMLR, 2023: 13584-13598.
>
> [2] Cui, B., Lupu, A., Sokota, S., Hu, H., Wu, D. J., & Foerster, J. N. (2023). Adversarial diversity in Hanabi. In The Eleventh International Conference on Learning Representations.
>
> [3] Rahman, M., Cui, J., & Stone, P. (2024, March). Minimum coverage sets for training robust ad hoc teamwork agents. In Proceedings of the AAAI Conference on Artificial Intelligence (Vol. 38, No. 16, pp. 17523-17530).
>
>
> ### Question
>
> > 1. This paper reminds me of [1]. Can the authors elaborate on how language conditioning and pre-strategy are related to each other?
>
> **Answer**: Thanks for your insightful viewpoint. First, [1] focused on how to implement human-AI coordination with LLMs as a medium between a human and AI agents, to guide human-AI coordination under human's instructions. In contrast, our work stands on the broader view on studying the principle to facilitate human-AI coordination with a medium as a indirect way to convey human's explicit goal. Therefore, [1] can be regarded as a good engineering work to further verify the validity of our theory on indirect human-AI interaction, where LLMs is an specification of the AI proxies equipped with pre-policies which can generate pre-strategies. As promising above, we have added this point in the related work.
>
> [1] Hu H, Sadigh D. Language instructed reinforcement learning for human-ai coordination[C]//International Conference on Machine Learning. PMLR, 2023: 13584-13598.
>
> > 2. How do you get the intrinsic rewards? How do humans specify their preferences?
>
> **Answer**: In our work, the intrinsic reward is manually crafted to encode human's desirable outcome (goal). However, the approach to represent human's desirable outcome is not the main gist of our paper as we mentioned above. We only provide a possible and initial solution. In contrast, [1] can be regarded as a kind of fancy approach to implement the algorithm we proposed, where intrinsic reward can be generated by LLMs to encode human's goal. We added an experiment to demonstrate the advantage of manually designing intrinsic rewards when it is feasible compared with the approach of learning an intrinsic reward in Appendix 18.

---

> > ### Comment · Reviewer_y3C4 · 2024-11-26
> >
> > I appreciate the author's response. I believe a human-crafted intrinsic/preference reward can not be generalized across environments; the interface to select NE should somewhat be automated to construct the preference to be more impressive for me to give a rating of 8. The analysis in MPE is more qualitatively like trajectory diversity (TrajDi) to me, probably because of the game's mechanism, which has less obvious NE than those games that have strong differences in NE (e.g. Hanabi H-Convention)
> > So, I am not motivated enough to adjust my score from 6 -> 8 given the above reasons right now.

---

> ### Author Response · Authors · 2024-11-26
>
> Thank you for your response! About the analysis of MPE, we agree that it is not a conventional case in games, but it is actually a case to show NE (since NE is just one solution concept to evaluate policies). The main reason for us to choose MPE is that this environment is more intuitive to most people with no prior knowledge of rules in Hanabi for visualization.
>
> Automated preference design can be treated as a further extension to replace the manually designed preference reward, but how to obtain intrinsic rewards is not our main contribution of this paper. In this paper, we primarily aim to justify the necessity of intrinsic rewards to express human's intention through the framework of MAIDs. In the future work, we may investigate how to automatically infer human's intention through additional data, when the human's intention is not able to be explicitly expressed.
>
> We would appreciate your further thoughts again and sincerely hope that you could reconsider your rating from the perspective of our main contributions. We would be highly appreciated if you could kindly reconsider to upgrade your overall rating, but we respect your decision if you choose to retain your current rating - thank you so much!

---

### Official Review · Reviewer_Dy2f · 2024-11-06

**Soundness:** 3
**Presentation:** 3
**Contribution:** 2
**Rating:** 5
**Confidence:** 3

**Summary:**

This paper introduces an innovative indirect human-AI interaction paradigm where humans do not directly engage with AI agents. Within this framework, the authors propose the concept of AI proxies that interact with other AI agents to achieve alignment with human-desired outcomes. The interactive processes are modeled using multi-agent influence diagrams, and the authors propose a pre-strategy intervention approach to achieve optimal Nash equilibrium in the game-theoretic setting. The effectiveness of this pre-strategy intervention is demonstrated through its integration into Multi-Agent Reinforcement Learning (MARL) and subsequent evaluation in two established MARL environments.

**Strengths:**

1. This paper studies an interest paradigm, i.e., the indirect human-AI interaction, which is common in the real world, such as remote surgery and mine rescue.
2. The paper demonstrates a well-structured and well-articulated presentation.
3. The introduction of experiment is complete and easy to follow.

**Weaknesses:**

1. The introduction of indirect human-AI interaction is not enough. From the Figure 1 (a)-(c), it cannot recognized which agents represent human, which represent the AI proxies, which represent the AI agents.
2. The author gives a proof about the existence of nash equilibrium in their game. However, they doesn't prove their proposed method, i.e., pre-strategy intervention based method can be ensured to converge to optimal nash equilibrium.
3. The selection of baseline in experiments are not proper. I suggest the authors choose some MARL algorithms, such as MAPPO and QMIX.

**Questions:**

1. Whether the introduction of AI proxies is rational? Whether AI proxies can also be regarded as AI agents? If yes, the definition of indirect interaction is not true.
2. Why the authors introduce LLM alignments in the related work. It seems that this research doesn't relevant to LLM alignment.
3. Whether the comprison experiments can be performed when pre-strategy intervention is not introduced into the approach.

**Details Of Ethics Concerns:**

No.

---

> ### Author Response · Authors · 2024-11-20
>
> We appreciate reviewer Dy2f's constructive feedback and positive comments on our idea is interest paradigm solve a very common real-world problem. We also thank for your recognize our paper is well-structured. We respond to each of the reviewer's questions in the following.
>
> ### Weaknesses
>
> > 1. The introduction of indirect human-AI interaction is not enough. From the Figure 1 (a-c) , it cannot recognized which agents represent human, which represent the AI proxies, which represent the AI agents.
>
> **Answer**: Thanks for rasing your concerns to help us improve our work. First, in Definition 1 we define that AI proxies are actually on behalf of a human. As a result, human would not appear in the IHAD. As for the differentiation between a human and AI proxies, this is our carelessness in writing. Actually, the color in red indicates AI proxies while the color in blue indicates AI agents. We have clarified this point in the revised paper.
>
> > 2. The author gives a proof about the existence of nash equilibrium in their game. However, they doesn't prove their proposed method, i.e., pre-strategy intervention based method can be ensured to converge to optimal nash equilibrium.
>
> **Answer**: Thanks for your critical viewpoint. We first agree that giving convergence result would make the paper more complete. Nevertheless, proving the convergence of an algorithm would need extensive technical assumptions and some further clarification. This would complicate the expression of our work and weaken the main contribution of our work at this moment, as our main purpose is to propose a new angle of constructing human-AI interaction based on graphical models (MAIDs). However, we agree that this is a good future direction to extend our work.
>
> > 3. The selection of baseline in experiments are not proper. I suggest the authors choose some MARL algorithms, such as MAPPO and QMIX.
>
> **Answer**: We agree that adding more baselines would strengthen the quality of our work. We have added new baselines, such as MAPPO, QMIX in Appendix 16.1 in the modified paper. As we see, our method can perform well on QMIX, but does not perform well on MAPPO. This sheds light on the design of intrinsic rewards could be not only corresponding to tasks, but also related to base algorithms. This is not surprising, as a base algorithm forms an optimization technique, which in combination with optimization problems as intrinsic rewards here, naturally together affect the final converggence result.
>
> ### Question
>
> > 1. Whether the introduction of AI proxies is rational? Whether AI proxies can also be regarded as AI agents? If yes, the definition of indirect interaction is not true.
>
> **Answer**: We understand your concerns. First, the introduction of AI proxies is reasonable, as in many cases a person is not able to interact with other AI agents (or other AI proxies on behalf other humans). In addition to the examples we gave in Introduction (related to physical constraints), we now show another example. Suppose a future scenario where a person named Bob need to interact with other AI agents (NPCs) and other AI proxies (can also be seen as AI agents to Bob in this case). Bob may not intend to expose himself to others because he would hide some private information (e.g. his facial profile). To satify this purpose, he has to hire an AI proxy.
>
> Second, as our above example shows, the definition of AI proxies and AI agents is always dependent on the ego view of a person of interest (e.g. Bob in the above example). Although from the system-wide view AI agents and AI proxies may be overlapped in definition as you observed, this is not what we claimed in Definition 1, where we always stand on the ego veiw of a human of interest. We have emphasized this point in the revised paper.
>
> Hope our answer has addressed your concerns.
>
> > 2. Why the authors introduce LLM alignments in the related work. It seems that this research doesn't relevant to LLM alignment.
>
> **Answer**: We agree with you and Reviewer rjtB that these two research fields have no connection on the surface. Thus, we have removed it in the revised paper.

---

> ### Comment · Reviewer_Dy2f · 2024-11-27
>
> Thank the authors for their response. Some key concerns still exist.
>
> 1. The response to my Question 3 was inappropriately placed within reviewer fGsJ's section without any cross-reference. The performance comparison in Figure 14 does not support the claim that "our proposed paradigm (with pre-policy) reaches the optimum more efficiently" as the difference appears negligible. The claim about handling "more complicated and realistic scenarios by separately designing pre-policy modules" seems to be questionable. In my experience, simpler approaches typically demonstrate better generalizability across different tasks.
>
> 2. Regarding Figure 1(a-c), there is no human role. I understand that if AI proxies are replaced by humans, the proposed algorithm would be applicable to Human-AI direct interaction scenarios, right?
>
> 3.  It is not convincing that the proposed approach performs worse than MAPPO, especially on an easy SMAC map. The individual reward setting is indeed important for algorithm performance. I suggest the authors should investigate and identify reward settings that effectively promote cooperation between AI proxies and AI agents.
>
> I will maintain my ratings. Thanks.

---

> ### Author Response · Authors · 2024-11-27
>
> 1. We sincerely apologize for the inconvenience caused by placing Question 3 in the wrong section. We agree that the claim "simpler approaches typically demonstrate better generalizability across different tasks" holds true in many scenarios. However, our method offers a novel perspective by addressing complex, realistic problems through the design of pre-policy mechanisms. The intention behind our pre-policy design is to tackle complex problems using simpler interaction strategies, which aligns with rather than contradicts this claim.
>
> 2. Yes, the graphical models we introduced in this paper can actually be extended to direct human-AI interaction scenarios. This can also be one significant novelty for our paper. In other words, albeit that the research problem is indirect human-AI interaction scenarios. The research method can also be employed to study more extensive problems. This can also be considered as one scientific contributions from our own perspectives.
>
> 3. We would like to kindly clarify that there might be some misunderstanding of our result. Although MAPPO can perform well in global rewards, **it does not satisfy the human desirable outcomes** (see Table 5). In summary, the main reason why the performance on task reward shown in Figure 13 is that our approach has to reach human's desirable outcomes (represented in intrinsic rewards) in addition to the task goal (represented in global rewards). This further justifies the difficulty to compromise human's desirable outcomes and the task goal, i.e., not every base algorithm can simply reach this balance, and thus the value of our research problem.
>
> We still thank you for your response and sincerely hope you can reconsider your decision, especially depending on the third point above.

---

### Official Review · Reviewer_fGsJ · 2024-11-10

**Soundness:** 3
**Presentation:** 3
**Contribution:** 3
**Rating:** 6
**Confidence:** 3

**Summary:**

This paper focuses on a problem scenario where humans interact with other AI agents via AI proxies. In this multi-agent environment with some specified group goal, the proxies are able to take into account human preferences via "pre-strategy intervention"

**Strengths:**

The motivation of intervening on multi-agent games through AI proxies is quite interesting because it allows us to see a mathematical strategy for interpolating between centralized and "independent" learning through the lens of intervening via proxies. Results are promising on a set of JaxMARL benchmarks.

**Weaknesses:**

The environments used for evaluation involve discrete actions. The formula to evaluate a pre-policy involves a summation over strategy profiles. How do you expect the results to scale up to more complex environments with continuous state and action spaces, including more complex observation spaces? There also isn't as much comparison of the paper's results to related works in inverse RL for multi-agent cooperation in the experiments section, although the ablations are helpful for understanding the author's strategy in developing their approach.

**Questions:**

Though cited in the main paper, I think it would be nice for the paper's presentation to put a motivating example of how MAIDs work in the main paper, rather than just the appendix. The paper could also benefit from a more extensive results discussion. I understand there is a lot of background to be laid out about causal models, etc., but I think these additions would actually help motivate and enhance readers' understanding of the work.

---

> ### Author Response · Authors · 2024-11-20
>
> We would like to thank reviewer fGsJ's feedback on our idea is well-motivated and interesting as well as our results are promising. We respond to each of the reviewer's questions in the sections below.
>
> ### Weaknesses
>
> > 1. The environments used for evaluation involve discrete actions. The formula to evaluate a pre-policy involves a summation over strategy profiles. How do you expect the results to scale up to more complex environments with continuous state and action spaces, including more complex observation spaces?
>
> **Answer**: Thanks for your insightful question. For continuous action spaces of pre-policy (pre-strategies), we may employ Monte Carlo sampling techniques or pick up a typical sample action. As for complicated continuous observation spaces, the key point to handle it is how to represent the observations in a compact form. We may suggest manually categorizing observations into several categories and then represent them via GNNs as we did in our first experiment. If the physical meaning of observations are obscure, we may employ some clustering techniques to acquire clusters from observations, e.g. the Principle Component Analysis (PCA). Although the above discussion is just prospect, we still directly evaluate our algorithm in a new environment in which observaitons are represented in more complex continuous values. The new experimental result is shown in Appendix 15 in the revised paper. In summary, our current method can still perform better than the baseline, but weakly. This indicates that we can improve this in the future research with the approeaches aforementioned.
>
> > 2. There also isn't as much comparison of the paper's results to related works in inverse RL for multi-agent cooperation in the experiments section, although the ablations are helpful for understanding the author's strategy in developing their approach.
>
> **Answer**: To our best knowledge, inverse RL is a algorithmic paradigm that learns a reward function from demonstration data and then the learned reward function can be used to train a policy. In contrast, in our paper we do not consider the case that demonstration data is provided, but we do not deny that this can be a future work to explore. Thanks for providing your viewpoint.
>
> ### Question
> > 1. Though cited in the main paper, I think it would be nice for the paper's presentation to put a motivating example of how MAIDs work in the main paper, rather than just the appendix.
>
> **Answer**: Thanks for your good point. We agree with you that moving the motivating example to the main paper could ease reading. However, we may have no extra pages to realize that. For this reason, giving an indicator to link to the MAID example could be a compromise.
>
> > 2. The paper could also benefit from a more extensive results discussion. I understand there is a lot of background to be laid out about causal models, etc., but I think these additions would actually help motivate and enhance readers' understanding of the work.
>
> **Answer**: If we understand you question correctly, we would say that we have involved detailed discussions on specific examples about pre-strategy interventions in Appendix 9 and the MPE experimental results in Appendix 11. If our understanding is incorrect, we look forward to your clarification and then give further responses.
>
> > 3. Whether the comprison experiments can be performed when pre-strategy intervention is not introduced into the approach.
>
> **Answer**: We have added the comparison in Appendix 16.2 in the revised paper. Since both paradigms are trained with intrinsic rewards, both of them finally reach the similar performance. However, our proposed paradigm (with pre-policy) reaches the optimum more efficiently. On the other hand, due to the flexibility of our proposed paradigm, it is potential to handle more complicated and realistic scenarios by separately designing pre-policy modules. This is what the paradigm without pre-policy cannot achieve.

---

> > ### Comment · Reviewer_fGsJ · 2024-11-25
> > **Response**
> >
> > These additions/responses from the author thoroughly address my concerns. I would raise soundness/presentation to 4, but keep my original overall rating of 6.

---

> ### Author Response · Authors · 2024-11-25
> **We are glad to find we have thoroughly addressed your concerns! Thank you for raising your soundness/presentation to 4!**
>
> Thank you for your timely response and raising your soundness/presentation score to 4! We are glad to found the additions/responses thoroughly address your concerns : ) We would be highly appreciated if you could kindly reconsider to upgrade your overall rating, but we respect your decision if you choose to retain your current rating - thank you so much!

---

> > ### Comment · Reviewer_fGsJ · 2024-11-25
> > **Response**
> >
> > My concerns and retaining my contribution rating echo the other reviewers comments about the "motivation for indirection human AI coordination" and use of AI proxies. I believe the scope of this motivation impedes my ability to rate the paper more highly given the lower potential impact of this contribution right now. Thus, I retain for now my original rating of marginal acceptance on the other merits of the paper.

---

### Author Response · Authors · 2024-11-20
**Appreciation to All Reviewers and Summary of the Revised Paper**

We sincerely appreciate all reviewers' great efforts on review and comments of our work. We thank for the positive comments:
- **Interesting, well-motivated and novel idea** (Reviewer fGsJ Dy2f rjtB)
- **Theoretical contributions** (Reviewer rjtB)
- **well-structured and well-articulated presentation** (Reviewer Dy2f)
- **Experiment easy to follow** (Reviewer Dy2f)

## Summary of Revised Paper
1. We added extra experiments on continuous space in Appendix 15 following Reviewer fGsJ's suggestions.
2. We added more experiments on QMIX and MAPPO as Reviewer Dy2f suggested in Appendix 16.
3. We added more experiments on comparing two-stage and simultaneous-update paradigm of training ProxyAgent as Reviewer rjtB suggested in Appendix 17.
4. We added a new experiment on comparing constructing intrinsic rewards manually to encode human's desirable outcomes, with an approach of learning intrinsic rewards to address Reviewer y3C4's concerns.
5. We refined the caption of Figure 1 as Reviewer Dy2f suggested.
6. We refined the definition of AI agents and AI proxies in Definition 1 as Reviewer Dy2f suggested at the bottom of Page 2.
7. We added the discussions about our work and two related works suggested by Reviewer rjtB and Reviewer y3C4.
8. We added the experiments on comparing our algorithm with its variant without pre-policy as Reviewer Dy2f suggested in Appendix 16.2.

**Note**: all added contents are highlighted in red.

---

### Author Response · Authors · 2024-11-25
**A Kind Reminder to All Reviewers**

Dear all reviewers,

Thanks for your efforts on reviewing our paper and giving insightful suggestions.

We believe we have tried our best efforts to address your questions and concerns, with adding a couple of new experiments and drafting answers.

As the time for discussion between authors and reviewers is coming to an end, we kindly remind you to take a look at our responses. If our responses have addressed your concerns, we kindly hope you can reconsider your ratings.

Best wishes

Authors

---

### Meta-Review · Area_Chair_RmFJ · 2024-12-19

**Metareview:**

The reviewers acknowledged that the paper tackles an interesting real-world setting of indirect AI-human interaction, and the proposed methodology to attain desirable human outcomes in such settings could be practically useful. However, the reviewers shared concern that the paper has limited contributions because the proposed methodology is not tailored to challenges of indirect interaction; moreover, there was a concern about a disconnect between theoretical/motivational justifications and the contributions from the proposed methodology as per the experiments. We want to thank the authors for their detailed responses. Based on the raised concerns and follow-up discussions, unfortunately, the final decision is a rejection. Nevertheless, this is exciting and potentially impactful work, and we encourage the authors to incorporate the reviewers' feedback when preparing a future revision of the paper.

**Additional Comments On Reviewer Discussion:**

The reviewers shared concern that the paper has limited contributions because the proposed methodology is not tailored to challenges of indirect interaction; moreover, there was a concern about a disconnect between theoretical/motivational justifications and the contributions from the proposed methodology as per the experiments. Even after the rebuttal phase, the concerns raised by reviewers remained. All the reviewers have borderline scores (5,5,6,6). Two out of four reviewers support a rejection decision with a clear justification as to why the paper is not yet ready for acceptance.

---

### Decision · Program_Chairs · 2025-01-22

Reject